

# Evolution of NO₃ reactivity during the oxidation of isoprene

Patrick Dewald[1], Jonathan M. Liebmann[1], Nils Friedrich[1], Justin Shenolikar[1], Jan Schuladen[1], Franz Rohrer[2], David Reimer[2], Ralf Tillmann[2], Anna Novelli[2], Changmin Cho[2], Kangming Xu[3], Rupert Holzinger[3], François Bernard[4,a], Li Zhou[4], Wahid Mellouki[4], Steven S. Brown[5,6], Hendrik Fuchs[2], Jos Lelieveld[1] and John N. Crowley[1]

[1]Atmospheric Chemistry Department, Max Planck Institut für Chemie, 55128 Mainz, Germany
[2]Institute of Energy and Climate Research, IEK-8: Troposphere, Forschungszentrum Jülich GmbH, 52428 Jülich, Germany
[3]Institute for Marine and Atmospheric Research, IMAU, Utrecht University, Utrecht, Netherlands
[4]Institut de Combustion, Aérothermique, Réactivité et Environnement (ICARE), CNRS (UPR 3021) /OSUC, 1C Avenue de la Recherche Scientifique, 45071 Orléans Cedex 2, France
[5]NOAA Chemical Sciences Laboratory, 325 Broadway, Boulder, CO 80305, USA
[6]Department of Chemistry, University of Colorado, Boulder, CO 80209, USA
[a]now at: Laboratoire de Physique et Chimie de l'Environnement et de l'Espace (LPC2E), Centre National de la Recherche Scientifique (CNRS), Université d'Orléans, Observatoire des Sciences de l'Univers en région Centre - Val de Loire (OSUC), Orléans, France

*Correspondence to*: John N. Crowley (john.crowley@mpic.de)

**Abstract.** In a series of experiments in an atmospheric simulation chamber (SAPHIR, Forschungszentrum Jülich, Germany) NO₃ reactivity ($k^{NO_3}$) resulting from the reaction of NO₃ with isoprene and stable trace gases formed as products was measured directly using a flow-tube reactor coupled to a cavity-ring-down spectrometer (FT-CRDS). The experiments were carried out in both dry and humid air with variation of the initial mixing ratios of ozone (50 – 100 ppbv), isoprene (3 – 22 ppbv) and NO₂ (5 – 30 ppbv). $k^{NO_3}$ was in excellent agreement with values calculated from the isoprene mixing ratio and the rate coefficient for the reaction of NO₃ with isoprene. This result serves both to confirm that the FT-CRDS returns accurate values of $k^{NO_3}$ even at elevated NO₂ concentrations and to show that reactions of NO₃ with stable reaction products like non-radical organic nitrates do not contribute significantly to NO₃ reactivity during the oxidation of isoprene. A comparison of $k^{NO_3}$ with NO₃ reactivities calculated from NO₃ mixing ratios and NO₃ production rates suggests that organic peroxy radicals and HO₂ account for ~ 50% of NO₃ losses. This contradicts predictions based on numerical simulations using the Master Chemical Mechanism (MCM version 3.3.1) unless the rate coefficient for reaction between NO₃ and isoprene-derived RO₂ is roughly doubled to ≈ $5 \times 10^{-12}$ cm³ molecule⁻¹ s⁻¹.

## 1 Introduction

The atmospheric oxidation of volatile organic compounds (VOCs) of both biogenic and anthropogenic origin has a great impact on tropospheric chemistry and global climate (Lelieveld et al., 2008). Isoprene is one of the major organic (non-methane) compounds that is released in the environment by vegetation and contributes ~ 50% to the overall emission of VOCs into the atmosphere (Guenther et al., 2012). The most important initiators of oxidation for biogenic VOCs in the atmosphere are





hydroxyl radicals (OH), ozone ($O_3$) and nitrate radicals ($NO_3$) (Geyer et al., 2001; Atkinson and Arey, 2003; Lelieveld et al.,

2016; Wennberg et al., 2018). Our focus in this study is on $NO_3$, which is formed via the sequential oxidation of NO by ozone (R1 and R2). During the daytime, $NO_3$ mixing ratios are very low owing to its efficient reaction with NO (R6) and its rapid photolysis (R7 and R8). Generally, $NO_3$ is present in mixing ratios greater than a few pptv only at night-time, when it can become the major oxidizing agent for VOCs including isoprene (R5). In forested regions, reactions with biogenic trace gases can however contribute significantly to the daytime reactivity of $NO_3$ (Liebmann et al., 2018a; Liebmann et al., 2018b).

Moreover, $NO_2$, $NO_3$ and $N_2O_5$ exist in thermal equilibrium (R3 and R4) so that the heterogeneous loss of $N_2O_5$ (and $NO_3$) at surfaces (R9 and R10) impacts on the lifetime of $NO_3$ in the atmosphere (Martinez et al., 2000; Brown et al., 2003; Brown et al., 2006; Brown et al., 2009b; Crowley et al., 2010).

| | |
|---|---|
| $NO + O_3 \rightarrow NO_2 + O_2$ | (R1) |
| $NO_2 + O_3 \rightarrow NO_3 + O_2$ | (R2) |
| 45    $NO_2 + NO_3 + M \rightarrow N_2O_5 + M$ | (R3) |
| $N_2O_5 + M \rightarrow NO_2 + NO_3 + M$ | (R4) |
| $NO_3 + isoprene \rightarrow products$ (e.g. $RONO_2$, $RO_2$) | (R5) |
| $NO_3 + NO \rightarrow 2NO_2$ | (R6) |
| $NO_3 + hv \rightarrow NO + O_2$ | (R7) |
| 50    $NO_3 + hv \rightarrow NO_2 + O$ | (R8) |
| $N_2O_5 + surface \rightarrow products$ (e.g. $HNO_3$) | (R9) |
| $NO_3 + surface \rightarrow products$ (e.g. particle nitrate) | (R10) |
| $RONO_2 + surface \rightarrow products$ (e.g. $HNO_3$) | (R11) |

Although isoprene is mainly emitted by vegetation at daytime (Sharkey and Yeh, 2001; Guenther et al., 2012), during which

its main sink reaction is with the OH radical (Paulot et al., 2012), it accumulates in the nocturnal boundary layer (Warneke et al., 2004; Brown et al., 2009a) where reactions of $NO_3$ and $O_3$ determine its lifetime (Wayne et al., 1991; Brown and Stutz, 2012; Wennberg et al., 2018). The rate constant (at 298 K) for the reaction between isoprene and $NO_3$ is $6.5 \times 10^{-13}$ cm$^3$ molecule$^{-1}$ s$^{-1}$, which is several orders of magnitude larger than for the reaction with $O_3$ ($1.28 \times 10^{-17}$ cm$^3$ molecule$^{-1}$ s$^{-1}$) (Atkinson et al., 2006; IUPAC, 2019) thus compensating for the difference in mixing ratios of $NO_3$ (typically 1-100 pptv) and

$O_3$ (typically 20-80 ppbv) (Edwards et al., 2017). $NO_3$ is often the most important nocturnal oxidant of biogenic VOCs (Mogensen et al., 2015) especially in remote, forested environments where it reacts almost exclusively with biogenic isoprene and terpenes (Ng et al., 2017; Liebmann et al., 2018a; Liebmann et al., 2018b). The reaction between isoprene and $NO_3$ leads initially to the formation of nitro isoprene peroxy radicals  (NISOPOO, e.g. $O_2NOCH_2C(CH_3)=CHCH_2OO$) that can either react with $NO_3$ forming mostly a nitro isoprene aldehyde (NC4CHO, e.g. $O_2NOCH_2C(CH_3)=CHCHO$) and methyl vinyl

ketone (MVK) or react further with other organic peroxy ($RO_2$) or hydroperoxy ($HO_2$) radicals forming nitrated carbonyls, peroxides and alcohols (Schwantes et al., 2015).



The organic nitrates formed ($RONO_2$) can deposit on particles (R11) and therefore the $NO_3$ + isoprene system contributes to the formation of secondary organic aerosols (SOA) (Rollins et al., 2009; Fry et al., 2018). Together with heterogeneous uptake of $N_2O_5$ or $NO_3$ on particle surfaces (R9 and R10), the build-up of SOA from isoprene oxidation products forms a significant

pathway for removal of reactive nitrogen species ($NO_x$) from the gas phase; a detailed understanding of the reaction between isoprene and $NO_3$ is therefore crucial for assessing its impact on SOA formation and $NO_x$ lifetimes.

In this study, the $NO_3$-induced oxidation of isoprene was examined in an environmental chamber equipped with a large suite of instruments including a cavity-ring-down spectrometer coupled to a flow-tube reactor (FT-CRDS) for direct $NO_3$ reactivity measurement (Liebmann et al., 2017). The $NO_3$ lifetime in steady-state (the inverse of its overall reactivity) has often been

derived from $NO_3$ mixing ratios and production rates, the latter depending on the mixing ratios of $NO_2$ and $O_3$ (Heintz et al., 1996; Geyer and Platt, 2002; Brown et al., 2004; Sobanski et al., 2016b). The steady-state approach works only if $NO_3$ is present at sufficiently large mixing ratios to be measured (generally not the case during daytime), breaks down to a varying extent if steady state is not achieved (Brown et al., 2003; Sobanski et al., 2016b) and may be influenced by heterogeneous losses of $NO_3$ or $N_2O_5$ (Crowley et al., 2011; Phillips et al., 2016) which are difficult to constrain. Comparing the steady-state

calculations with the FT-CRDS approach (which derives the $NO_3$ reactivity attributable exclusively to VOCs) can provide insight into the main contributions to $NO_3$ reactivity and its evolution as the reaction progresses. In the following, we present the results of direct $NO_3$ reactivity measurements in the SAPHIR environmental chamber under controlled conditions and explore the contributions of isoprene, peroxy radicals and stable oxidation products to $NO_3$ reactivity over a period of several hours as the chemical system resulting from $NO_3$ induced oxidation of isoprene evolves.

**2 Measurement and instrumentation**

An intensive study of the $NO_3$ + isoprene system (NO3ISOP campaign) took place at the SAPHIR chamber of the Forschungszentrum Jülich over a three-week period in August 2018. The aim of NO3ISOP was to improve our understanding of product formation in the reaction between $NO_3$ and isoprene as well as its impact on the formation of secondary organic aerosols (SOA). Depending on the conditions (high or low $HO_2$/$RO_2$, temperature, humidity, daytime or night-time) a large

variety of oxidation products, formed via different reaction paths exist (Wennberg et al., 2018). During NO3ISOP, the impact of varying experimental conditions on the formation of gas phase products as well as secondary organic aerosol formation and composition was explored within 22 different experiments (see Table 1). Typical conditions were close to those found in the atmosphere with 5 ppbv of $NO_2$, 50-100 ppbv of $O_3$ and 3 ppbv of isoprene or (when high product formation rates were required) the $NO_2$ was raised to 25 ppbv and isoprene to 10 ppbv. The high $O_3$ mixing ratios in the chamber ensured that NO

was not detectable (< 10 pptv) in the darkened chamber.

The first 11 experiments of the NO3ISOP were dedicated to gas-phase chemistry; in the second part seed-aerosol (($NH_4$)$_2$$SO_4$) was added and the focus shifted to aerosol measurements. Due to a contamination event in the chamber the experiment from



the 7th August is not considered for further analysis. The SAPHIR chamber and the measurements/instruments that are relevant for the present analysis are described briefly below.

## 2.1 The SAPHIR chamber

The atmospheric simulation chamber SAPHIR has been described in detail on various occasions (Rohrer et al., 2005; Bossmeyer et al., 2006; Fuchs et al., 2010) and we present only a brief description of some important features here: The outdoor chamber consists of two layers of FEP foil defining a cylindrical shape with a volume of 270 m³ and a surface area of 320 m². The chamber is operated at ambient temperature and its pressure is ~30 Pa above ambient level. A shutter system in the roof enables the chamber to be completely darkened or illuminated with natural sunlight. Two fans result in rapid (2 min) mixing of the gases in the chamber, which was flushed with 250 m³ h$^{-1}$ of synthetic air (obtained from mixing high purity nitrogen and oxygen) for several hours between each experiment. Leakages and air consumption by instruments leads to a dilution rate of typically 1.4 x 10$^{-5}$ s$^{-1}$. Coupling to a separate plant chamber enabled the introduction of plant emissions into the main chamber (Hohaus et al., 2016).

## 2.2 NO₃ reactivity measurements: FT-CRDS

The FT-CRDS instrument for directly measuring NO$_3$ reactivity ($k^{NO_3}$) has been described in detail (Liebmann et al., 2017) and only a brief summary is given here. NO$_3$ radicals are generated in sequential oxidation of NO with O$_3$ (reactions R1 and R2) in a darkened, thermostated glass reactor at a pressure of 1.3 bar. The reactor surfaces are coated with Teflon (DuPont, FEPD 121) to reduce the loss of NO$_3$ and N$_2$O$_5$ at the surface during the ~ 5 min residence time. The gas mixture exiting the reactor (400 sccm) is heated to 140°C before being mixed with either zero-air or ambient air (at room temperature) and entering the FEP-coated flow-tube where further NO$_3$ production (R2), equilibrium reaction with N$_2$O$_5$ (R3 and R4) as well as NO$_3$ loss via reactions with VOCs/NO (R5/R6) or with the reactor wall (R10) take place. NO$_3$ surviving the flow reactor after a residence time of 10.5 s is quantified by CRDS at a wavelength of 662 nm. The NO$_3$ reactivity is calculated from relative change in NO$_3$ concentration when mixed with zero-air or ambient air. In order to remove a potential bias by ambient NO$_3$/N$_2$O$_5$, sampled air is passed through an uncoated 2L glass flask (~60 s residence time) heated to 45°C to favour N$_2$O$_5$ decomposition before reaching the flowtube. Ambient NO$_3$ (or other radicals, e.g RO$_2$) is lost by its reaction with the glass walls. In addition to the reaction of interest (R5), reactions (R2) to (R4) and (R10) affect the measured NO$_3$ concentration so that corrections via numerical simulation of this set of reactions are necessary to extract $k^{NO_3}$ from the measured change in NO$_3$ concentration, necessitating accurate measurement of O$_3$, NO and especially NO$_2$ mixing ratios. For this reason, the experimental setup was equipped with a second cavity for the measurement of NO$_2$ at 405 nm as described recently (Liebmann et al., 2018b). In its current state the instrument's detection limit is ~ 0.005 s$^{-1}$. By diluting highly reactive ambient air with synthetic air, ambient reactivities up to 45 s$^{-1}$ can be measured. The overall uncertainty in $k^{NO_3}$ results from instability of the NO$_3$ source and the CRDS detection of NO$_3$ and NO$_2$ as well as uncertainty introduced by the numerical simulations. Under laboratory conditions, measurement errors result in an uncertainty of 16%. The uncertainty associated with the numerical simulation was estimated



by Liebmann et al. (2017) who used evaluated rate coefficients and associated uncertainties (IUPAC) to show that the uncertainty in $k^{NO_3}$ is highly dependent on the ratio between the $NO_2$ mixing ratio and the measured reactivity. If a reactivity of 0.046 s⁻¹ (e.g. from 3 ppbv of isoprene), is measured at 5 ppbv of $NO_2$ (typical for this campaign), the correction derived from the simulation would contribute an uncertainty of 32% to the resulting overall uncertainty of 36%. For an experiment with 25 ppbv of $NO_2$ and 10 ppbv of isoprene, large uncertainties ($> 100\%$) are associated with the correction procedure as the

$NO_3$ loss caused by reaction with $NO_2$ exceeds VOC-induced losses. Later we show that data obtained even under unfavorable conditions (high $NO_2$ mixing ratios) are in accord with isoprene measurements, which suggests that the recommended uncertainties in rate coefficients for R3 and R4 are overly conservative.

The sampled air was typically mixed with ~50 pptv of $NO_3$ radicals and the reaction between $NO_3$ and $RO_2$ radicals generated in the flow-tube (R5) represents a potential bias to the measurement of $k^{NO_3}$. In a typical experiment (e.g. 3 ppbv of isoprene)

the reactivity of $NO_3$ towards isoprene is 0.046 s⁻¹. A simple calculation shows that a total of 20 pptv of $RO_2$ radicals have been formed after 10.5 s reaction between $NO_3$ and isoprene time in the flow tube. Assuming a rate coefficient of ~$5 \times 10^{-12}$ cm³ molecule⁻¹ s⁻¹ for reaction between $NO_3$ and $RO_2$, we calculate a 5% contribution of $RO_2$ radicals to $NO_3$ loss. In reality, this value represents a very conservative upper limit as $RO_2$ is present at lower concentrations throughout most of the flow tube and its concentration will be significantly reduced by losses to the reactor wall and self-reaction. In our further analysis

we therefore do not consider this reaction.

### 2.3 VOC measurements: PTR-ToF-MS

During the NO3ISOP campaign, isoprene and other VOCs were measured by two different PTR-ToF-MS (Proton Transfer Reaction Time-Of-Flight Mass Spectrometer) instruments. The PTR-TOF1000 (IONICON Analytic GmbH) has a mass resolution > 1500 m/Δm and a limit of detection <10 ppt for a 1 minute integration time. The instrumental background was

determined every hour by pulling the sample air through a heated tube (350˚C) filled with a Pt catalyst for 10 minutes. Data processing was done using PTRwid (Holzinger, 2015) and the quantification/calibration was done following the procedure as described recently (Holzinger et al., 2019).

The Vocus PTR (Tofwerk AG/Aerodyne Research Inc.) features a newly designed focusing ion-molecule reactor resulting in a resolving power of 12000 m/Δm (Krechmer et al., 2018). The isoprene measurements of the two instruments agreed within

the uncertainties. For the evaluation of the experiment on the 2ⁿᵈ August only data from the PTR-TOF1000 were available. For all the other experiments of the campaign, isoprene and monoterpene mixing ratios were taken from the Vocus PTR owing to its higher resolution and data coverage.

### 2.4 NO₃/N₂O₅/NO₂/NO/O₃ measurements

The $NO_3$/$N_2O_5$ mixing ratios used for analysis are from a harmonized data set including the measurements from two CRDS

instruments. Data availability, quality and consistency with the expected $NO_3$ / $N_2O_5$ / $NO_2$ equilibrium ratios were criteria for selecting which data set to use for each experiment. Both instruments measure $NO_3$ (and $N_2O_5$ after its thermal decomposition





to NO$_3$ in a heated channel) using cavity ring down spectroscopy at a wavelength of ~662 nm. The 5-channel device operated by the Max-Planck-Institute (MPI) additionally measured NO$_2$ and has been described recently in detail (Sobanski et al., 2016a). Its NO$_3$ channel has a limit of detection (LOD) of 1.5 pptv (total uncertainty of 25%); the N$_2$O$_5$ channel has a LOD of

3.5 pptv (total uncertainty of 28% for mixing ratios between 50 and 500 pptv). Air was sub-sampled from a bypass flow drawing ~40 SLM through a 4m length of 0.5 inch (inner diameter, i.d.) PFA tubing from the chamber. Variation of the bypass flow rate was used to assess losses of NO$_3$ (< 10%) in transport to the instrument, for which correction was applied. Air entering the instrument was passed through a Teflon membrane filter (Pall Corp., 47mm, 0.2 μm pore) which was changed every 60 mins. Corrections for loss of NO$_3$ and N$_2$O$_5$ on the filter and inlet lines were carried out as described previously (Sobanski et

al., 2016a).

The second CRDS was built by the NOAA Chemical Sciences Laboratory (Dubé et al., 2006; Fuchs et al., 2008; Wagner et al., 2011; Fuchs et al., 2012; Dorn et al., 2013) and was operated by the Institut de Combustion, Aérothermique, Réactivité et Environnement (ICARE). During the NO3ISOP campaign, the NOAA-CRDS was positioned beneath the chamber and air was sampled through an individual port in the floor. The sampling flow rate was 5.5-7 L min$^{-1}$ through a Teflon FEP line (i.d. 1.5

mm, total length about 0.9 m) extending by about 50 cm (i.d. 4 mm) with 25 cm (i.d. 4 mm) into the chamber. A Teflon filter (25 μm thickness, 47 mm diameter, 1-2 μm pore size) was placed downstream of the inlet to remove aerosol particles, and changed automatically at an interval of 1.5 - 2 h depending on the conditions of the experiments, such as the amount of aerosol in the chamber. The instrument was operated with a noise equivalent 1σ detection limit of 0.25 and 0.9 pptv in 1s for the NO$_3$ and N$_2$O$_5$ channels, respectively. The total uncertainties (1σ) of the NOAA-CRDS instrument were 25% (NO$_3$) and -8%/+11%

(N$_2$O$_5$).

NO$_2$ mixing ratios were taken from a harmonized data set combining the measurements of the 5-channel CRDS with that of the NO$_3$ reactivity setup as well as the NO$_x$ measurement of a thermal dissociation CRDS setup (Thieser et al., 2016). The NO$_x$ measurement could be considered as a NO$_2$ measurement since during dark periods of the experiments NO would have been present at extremely low levels. The total uncertainty associated with the NO$_2$ mixing ratios is 9%.

NO was measured with an LOD of 4 pptv via chemiluminescence (CL; (Ridley et al., 1992)) detection (ECO Physics, model TR780) and ozone was quantified with an LOD of 1 ppbv by ultraviolet absorption spectroscopy at 254 nm (Ansyco, ozone analyser 41M). Both instruments operate with an accuracy (1σ) of 5%.

## 2.5 Box model

The results of the chamber experiments were analysed using a box model based on the oxidation of isoprene by NO$_3$, OH and

O$_3$ as incorporated in the Master Chemical Mechanism (MCM), version 3.3.1 (Saunders et al., 2003; Jenkin et al., 2015). In this work, the analysis focusses on the fate of the NO$_3$ radical, so that the oxidation of some minor products was omitted in order to reduce computation time. Moreover, the most recently recommended rate coefficient (IUPAC, 2019) for the reaction between NO$_3$ and isoprene ($k_5 = 2.95 \times 10^{-12}$ exp(-450/$T$) cm$^3$ molecule$^{-1}$ s$^{-1}$) was used instead of the value found in the MCM v3.3.1, which is 6.8% higher. Chamber-specific parameters such as temperature, pressure as well as the time of injection and



amount of trace gases added (usually $O_3$, $NO_2$ and isoprene) were the only constraints to the model. The chamber dilution flow was implemented as first-order loss rates for all trace-gases and wall loss rates for $NO_3$ or $N_2O_5$ were introduced (see Section 3.2). The numerical simulations were performed with FACSIMILE/CHEKMAT (release H010 date 28 April 1987 version 1) at 1 minute time resolution (Curtis and Sweetenham, 1987). The chemical scheme used is listed in the supplementary information (Table S1).

## 3 Results and discussion

An overview of the experimental conditions (e.g. isoprene, $NO_3$, $NO_2$ and $O_3$ mixing ratios) on each day of the campaign is given in Fig. 1. The temperature in the chamber was typically between 20 and 30 °C but increased up to 40 °C when the chamber was opened to sunlight. The relative humidity was close to 0% during most of the experiments before $14^{th}$ August. After this date, the experiments focussed on secondary organic aerosol formation and humidified air was used.

We divide the experiments into two broad categories according to the initial conditions: Type 1 experiments were undertaken with $NO_3$ production from 5 ppbv of $NO_2$ and 100 ppbv of $O_3$. The addition of isoprene with mixing ratios of ~3 ppbv resulted in $NO_3$ reactivities of around 0.05 $s^{-1}$ at the time of injection. The $NO_3$ and $N_2O_5$ mixing ratios were typically of the order of several tens of pptv in the presence of isoprene under dry conditions. During humid experiments (with seed aerosol) $NO_3$ mixing ratios were mostly below the LOD in the presence of isoprene owing to increased uptake of $NO_3$/$N_2O_5$ on particles.

An exceptionally large isoprene injection (~20 ppbv) resulted in the maximum $NO_3$ reactivity of 0.4 $s^{-1}$ on the $24^{th}$ August. In type 2 experiments, higher $NO_3$ production rates were achieved by using 25 ppbv of $NO_2$ and 100 ppbv of $O_3$. In these experiments, with the goal of generating high concentrations of organic oxidation products, isoprene mixing ratios of 10 ppbv resulted in reactivities of ~0.2 $s^{-1}$ at the time of isoprene injection. Owing to high $NO_3$ production rates, several hundreds of pptv of $NO_3$ and a few ppbv of $N_2O_5$ were present in the chamber.

Figure 1 shows that once isoprene has been fully removed at the end of each experiment, the $NO_3$ reactivity tends towards its LOD of 0.005 $s^{-1}$ indicating that the evolution of the $NO_3$ reactivity is closely linked to the changing isoprene mixing ratio.

### 3.1 Comparison of $k^{NO_3}$ with calculated reactivity based on measurements of VOCs

The VOC contribution to the $NO_3$ reactivity is the summed, first-order loss rate coefficient attributed to all VOCs present in the chamber according to Eq. (1):

$$k^{NO_3} = \sum k_i [VOC]_i \qquad (1)$$

where $k_i$ is the rate coefficient (cm$^3$ molecule$^{-1}$ s$^{-1}$) for the reaction between a VOC of concentration $[VOC]_i$ and $NO_3$.

Reliable values of $k^{NO_3}$ and VOC data are available from the $2^{nd}$ of August onwards (see Table 1 for experimental conditions) and were used to compare FT-CRDS measurements of $k^{NO_3}$ with $\sum k_i[VOC]_i$. For most of the experiments, isoprene was the only VOC initially present in the chamber and at the beginning of the experiments $k^{NO_3}$ should be given by $k_5$[isoprene], the

latter measured by the PTR-MS instruments (see above). On the $9^{th}$ and $21^{st}$ August, both isoprene and propene (100 ppbv)



were injected into the chamber, the summed $NO_3$ reactivity from these trace gases was then: $k_5$[isoprene]+$k_{propene}$[propene], with $k_{propene}$ = 9.5 x $10^{-15}$ $cm^3$ molecule$^{-1}$ s$^{-1}$ at 298 K (IUPAC, 2019). As no propene data was available, the propene mixing ratios were assessed with the model (see above) based on injected amounts as well as subsequent loss by oxidation chemistry (mainly ozonolysis) and dilution. On the 22$^{nd}$ August, coupling to a plant emission chamber permitted the introduction of

monoterpenes and isoprene into the main chamber so that the $NO_3$ reactivity was $k_5$[isoprene]+$k_{monoterpenes}$[monoterpenes]. The uncertainty in $\Sigma k_i$[VOC]$_i$ was propagated from the standard deviation of the isoprene and monoterpene mixing ratios and the uncertainties of 41% in $k_5$, 58% in $k_{propene}$ (IUPAC, 2019) as well as 47% in $k_{monoterpenes}$ (average uncertainty of three dominant terpenes, see below).

Figure 2 (a) depicts an exemplary time series of $k^{NO_3}$ and $\Sigma k_i$[VOC]$_i$ between the 9$^{th}$ and 13$^{th}$ of August. The measured $k^{NO_3}$

and values of $\Sigma k_i$[VOC]$_i$ calculated from measured isoprene (and modelled propene in case of the 9$^{th}$ August) are, within experimental uncertainty, equivalent indicating that the $NO_3$ reactivity can be attributed entirely to its reaction with isoprene (and other reactive trace gases like propene) injected into the chamber.

The correlation between $k^{NO_3}$ and $\Sigma k_i$[VOC]$_i$ for the entire campaign dataset is illustrated in Fig. 2(b). Type 2 experiments (high $NO_2$ mixing ratios) were included despite the unfavourable conditions for measurement of $k^{NO_3}$, which result in large

correction factors via numerical simulation (see above). The data points obtained on the 14$^{th}$ August display large variability, which is likely to have been caused by non-operation of the fans leading to poor mixing in the chamber. An unweighted linear regression of the whole dataset yields a slope of 0.962 ± 0.003 indicating excellent agreement between the directly measured $NO_3$ and those calculated from Eq. (1). The intercept of (0.0023 ± 0.0004) s$^{-1}$ is below the LOD of the reactivity measurement. Note that data from the 7$^{th}$ August (chamber contamination) were not used. On the 15$^{th}$ and 21$^{st}$ August, additional flushing of

the chamber with synthetic air (150-300 $m^3$) and humidification shortly before the actual beginning of the experiment resulted in a constant background reactivity in $k^{NO_3}$ of 0.04 s$^{-1}$ on the 15$^{th}$ and 0.012 s$^{-1}$ on the 21$^{st}$ August. High background reactivity was not observed during other humid experiments if the chamber was flushed extensively with synthetic air (~2000 $m^3$) during the night between experiments and if the additional flushing was omitted. The trace gas(es) causing this background reactivity could not be identified with the available measurements, but are probably released from the chamber walls during flushing and

humidification. In order to make detailed comparison with the VOC data the background reactivity, which was fairly constant, was simply added.

A more detailed examination of $k^{NO_3}$ data from two type 1 experiments (low $NO_2$) is given in Fig. 3. The grey shaded areas indicate the total uncertainty associated with the FT-CRDS measurement of $k^{NO_3}$ (Liebmann et al., 2017), the scatter in the data stems mostly from the correction procedure via numerical simulation.

On the 20$^{th}$ August (upper panel, Fig. 3a) in addition to $NO_2$ and $O_3$, $(NH_4)_2SO_4$ seed aerosol (~50 µg $cm^{-3}$) and β-caryophyllene (~ 2 ppbv) were injected at 08:40 UTC in order to favour formation of secondary organic aerosol. The presence of β-caryophyllene explains the small increase in the $NO_3$ reactivity after 08:30 UTC. As the lifetime of β-caryophyllene is extremely short in the chamber under the given conditions, its contribution to $k^{NO_3}$ was short-lasting. At 09:20, 10:13 and



11:50 UTC isoprene was injected into the chamber resulting in step-like increases in the measured $NO_3$ reactivity. Each
increase in reactivity and the ensuing evolution over time match well with the calculated values of $k_5$[isoprene] (red datapoints).
The red shaded area indicates the overall uncertainty in the latter. Clearly, within experimental uncertainty, the $NO_3$ reactivity
is driven almost entirely by reaction with isoprene, with negligible contribution from stable, secondary products.

During the experiment of the 23$^{rd}$ August (lower panel, Fig. 3b), only isoprene and ozone were present in the chamber for the
first 4 hours. The absence of $NO_2$ results in a more accurate, less scattered measurement of $k^{NO_3}$ and underscores the reliability
of the measurement under favourable conditions. All of the observed reactivity can be assigned to isoprene that was injected
at 06:52 UTC.

The results of a type 2 experiment with $NO_2$ mixing ratios of ~ 20 ppbv as well as higher isoprene mixing ratios (injections of
~8 and ~3 ppbv under dry conditions) is depicted in Fig. 4 (a). Despite the requirement of large correction factors to $k^{NO_3}$
owing to the high $NO_2$ to isoprene ratios, fair agreement between measured $k^{NO_3}$ and the expected reactivity is observed for
each of the isoprene injections at 07:30, 09:20 and 10:50 UTC. The agreement may indicate that the uncertainty in $k^{NO_3}$ (grey
shaded area) which is based on uncertainty in e.g. the rate coefficient for reaction between $NO_3$ and $NO_2$ (Liebmann et al.,
2017) is overestimated.

In Fig. 4(b) we display the results of an experiment on 12$^{th}$ August, in which the initially darkened chamber (first ~4 hours)
was opened to sunlight (final 4 hours). $NO_2$ mixing ratios varied between 12 and 4 ppbv and isoprene was injected (~3 ppbv)
three times at 05:55, 07:40 and 09:45 UTC. During the dark-phase, measured $k^{NO_3}$ follows $k_5$[isoprene]. At 11:00 UTC the
chamber was opened to sunlight, during which, approximately 5 ppbv of $NO_2$, 200 – 150 pptv of NO and < 1 ppbv of isoprene
were present in the chamber. In this phase, the loss of $NO_3$ was dominated by its photolysis and reaction with NO. Within
experimental uncertainty, the measured daytime $k^{NO_3}$ after correction for both $NO_2$ and NO (correction factors between 0.05
and 0.02) during the sunlit period was still close to $k_5$[isoprene].

On the 22$^{nd}$ August, the SAPHIR chamber was filled with air from a plant chamber (SAPHIR-PLUS) containing six European
oaks (*Quercus robur*) which emit predominantly isoprene but also monoterpenes, mainly limonene, 3-carene and α-pinene
(van Meeningen et al., 2016).

The time series of measured $NO_3$ reactivity $k^{NO_3}$ (black datapoints) after coupling to the plant-chamber at 08:00 UTC is shown
in Fig. 5. Data after 11:40 UTC is not considered as the chamber lost its pressure after several re-coupling attempts to the plant
chamber. Also plotted (red data points) is the $NO_3$-reactivity calculated from $\Sigma k_i$[VOC]$_i$ whereby both isoprene and the total
terpene mixing ratio (up to 500 pptv) were measured by the Vocus PTR-MS. As only the mixing ratio of the sum of the
monoterpenes was known, an average value of the very similar $NO_3$ rate coefficients (IUPAC, 2019) for limonene, 3-carene
and α-pinene was used for the calculation of $\Sigma k_i$[VOC]$_i$ with $k_{monoterpenes}$ = 9.1 x 10$^{-12}$ cm$^3$molecule$^{-1}$s$^{-1}$ (analogously averaged
uncertainty of 47%). Figure 5 indicates very good agreement between measured and calculated $NO_3$ reactivity, with ~ 70% of
the overall reactivity caused by isoprene, which is indicated by the purple, shaded area. Despite being present at much lower
mixing ratios that isoprene, the terpenes contribute ~ 30% to the overall $NO_3$ reactivity, which reflects the large rate constants
for reaction of $NO_3$ with terpenes.





The experiments described above indicate that, for a chemical system containing initially only isoprene as the reactive organic trace gas, the measured values of $k^{NO_3}$ can be fully assigned to the isoprene present in the chamber over the course of its

degradation. During the NO3ISOP campaign, not only NO₃ reactivity but also OH-reactivity ($k^{OH}$) was measured; the experimental technique is described briefly in the supplementary information. A detailed analysis of the OH-reactivity data-set will be subject of a further publication and in Fig. S1 we only compare values of $k^{NO_3}$ and $k^{OH}$ obtained directly after isoprene injections, where $k^{OH}$ should not be significantly influenced by the reaction of OH with secondary products. As shown in Fig. S1, isoprene concentrations derived from both $k^{NO_3}$ and $k^{OH}$ are generally in good agreement when [isoprene]

< 5 ppbv.

The oxidation of isoprene by NO₃ in air results in the formation of stable (non-radical) products as well as organic peroxy radicals ($RO_2$) that can also react with NO₃. As radicals (e.g. NO₃, $RO_2$ and $HO_2$) are not sampled by the FT-CRDS, the equivalence of $k^{NO_3}$ and $k_5$[isoprene] indicates that non-radical, secondary oxidation products do not contribute significantly to the NO₃ reactivity.

**3.2 Steady-state and model calculations: Role of RO₂ & chamber walls**

The contribution of RO₂, HO₂ and stable products to NO₃ reactivity was examined using a box-model based on the chemical mechanistic oxidation processes of isoprene by NO₃, OH and O₃ as incorporated in the Master Chemical Mechanism, version 3.3.1 (Saunders et al., 2003; Jenkin et al., 2015; Khan et al., 2015). A numerical simulation (Fig. 6) of the evolution of NO₃ reactivity was initialised using the experimental conditions of the first isoprene injection on 10[th] August (5.5 ppbv NO₂, 60

ppbv O₃ and 2 ppbv isoprene, dry air) including chamber-specific parameters such as temperature, the NO₃ and N₂O₅ wall loss rates (quantified in detail below) and the dilution rate. In the model, NO₃ reacts both with stable products and peroxy radicals. The major, stable oxidation product according to MCM is an organic nitrate with aldehyde functionality ($O_2NOC_4H_6CHO$, NC4CHO). As the corresponding rate coefficient for the reaction of this molecule with NO₃ is not known, MCM uses a generic rate coefficient based on the IUPAC-recommended, temperature-dependent expression for acetaldehyde + NO₃ scaled with a

factor of 4.25 to take differences in molecular structure into account. The maximum, modelled mixing ratio of NC4CHO was ~ 5 ppbv in type 2 experiments which would result in a NO₃ reactivity of 0.001 s⁻¹. This value is below the instrument's LOD and would only become observable at extremely low isoprene concentrations. As apparent in Fig. 6, the contribution of stable oxidation products (blue) to the NO₃ reactivity is insignificant compared to the primary oxidation of isoprene (red).

Since the rate coefficients for reaction of isoprene derived peroxy radicals and NO₃ are (unlike NO₃ + HO₂) poorly constrained

by experimental data, the MCM uses a generic value of $2.3 \times 10^{-12}$ cm³molecule⁻¹s⁻¹ which is based on rate coefficient for the reaction between NO₃ and $C_2H_5O_2$. The modelled, overall NO₃ reactivity when reactions with RO₂ and HO₂ are included (black line) is on average 22% higher than the reactivity associated only with isoprene, the major contributors to the additional NO₃ reactivity being nitrooxy isopropyl peroxy radicals ($O_2NOC_5H_8O_2$, NISOPOO) formed in the primary oxidation step. As neither RO₂ nor HO₂ radicals will survive the inlet tubing (and heated glass flask) between the SAPHIR chamber and the FT-

CRDS instrument, our measurement of $k^{NO_3}$ does not include their contribution. The measured values of $k^{NO_3}$ (black





datapoints) scatter around the isoprene-induced reactivity (red) which is understood to result from the minor role of stable (non-radical) oxidation products (blue) in removing $NO_3$ and the exclusion of peroxy radicals in the measurement.

Another method of deriving $NO_3$ reactivity is to calculate it from $NO_3$ (and/or $N_2O_5$) mixing ratios and production rates under the assumption of steady-state as has been carried out on several occasions for the analysis of ambient $NO_3$ measurements

(Heintz et al., 1996; Geyer and Platt, 2002; Brown et al., 2004; Sobanski et al., 2016b). In contrast to our direct measurement of $k^{NO_3}$, all loss processes (including reaction of $NO_3$ with $RO_2$, $HO_2$ and uptake of $NO_3$ and $N_2O_5$ to surfaces) are assessed using the steady-state calculations. A comparison between $k^{NO_3}$ and $NO_3$ reactivity based on a steady-state analysis should enable us to extract the contribution of peroxy radicals and wall-losses of $NO_3$ in the SAPHIR chamber. In steady-state, the $NO_3$ reactivity ($k_{ss}^{NO_3}$) is derived from the ratio between the $NO_3$-production rate via reaction (R2) with rate coefficient $k_2$ and

the mixing ratios of $O_3$, $NO_2$ and $NO_3$ (Eq.2).

$$k_{ss}^{NO_3} = \frac{k_2[O_3][NO_2]}{[NO_3]} \tag{2}$$

Acquiring steady-state can take several hours if the $NO_3$ lifetime is long, temperatures are low or $NO_2$ mixing ratios are high (Brown et al., 2003). In the NO3ISOP experiments, the $NO_3$ reactivities were generally high, and steady-state is achieved within a few minutes of isoprene being injected into the chamber. However, $NO_2$ re-injections in the chamber during periods

of low reactivity at the end of an experiment when isoprene was already depleted can lead to a temporary breakdown of the steady-state assumption. In order to circumvent this potential source of error the non-steady-state reactivities ($k_{nss}^{NO_3}$) based on $NO_3$ and $N_2O_5$ measurements (McLaren et al., 2010) were calculated using Eq. (3).

$$k_{nss}^{NO_3} = \frac{k_2[O_3][NO_2] - \frac{d[NO_3]}{dt} - \frac{d[N_2O_5]}{dt}}{[NO_3]} \tag{3}$$

This expression is similar to Eq. (2) except for the subtraction of the derivatives $d[NO_3]/dt$ and $d[N_2O_5]/dt$ from the production

term. A comparison of $k_{ss}^{NO_3}$ and $k_{nss}^{NO_3}$ is given in the SI and verifies the assumptions above: As soon as isoprene is injected into the system $k_{ss}^{NO_3}$ and $k_{nss}^{NO_3}$ are equivalent (see Fig. S2a) but $k_{ss}^{NO_3}$ shows short-term deviations at $NO_2$ reinjections (see Fig. S2b). As the non-steady-state reactivities are less affected by such events, the latter were used for the comparison with the measured $NO_3$ reactivities. The steady-state as well as the non-steady-state calculations are only valid if equilibrium between $NO_3$ and $N_2O_5$ is established. Moreover, the $N_2O_5$ measurements are usually less sensitive to instrument-specific losses under

dry conditions. For this reason, measured $NO_3$ mixing ratios were checked for consistency with the equilibrium to $N_2O_5$ using the equilibrium constant $K_{eq}$ for reactions (R3)/(R4) as well as the measured $N_2O_5$ and $NO_2$ mixing ratios as denoted in Eq. (4) for this analysis.

$$[NO_3]_{eq} = \frac{[N_2O_5]}{K_{eq}[NO_2]} \tag{4}$$

A time series of measured $k^{NO_3}$ and calculated $k_{nss}^{NO_3}$ is depicted in Fig. 7a, which shows the results from experiments in the

absence of aerosol only. It is evident that $k_{nss}^{NO_3}$ is much higher than $k^{NO_3}$. In Fig. 7b we plot $k^{NO_3}$ versus $k_{nss}^{NO_3}$: An unweighted, orthogonal, linear fit has a slope of $0.54 \pm 0.01$ and indicates that the measured values of $k^{NO_3}$ are almost a factor of two lower





than $k_{nss}^{NO_3}$. Propagation of the uncertainties in $k_2$ (15%; IUPAC, 2019) and the NO$_3$ , NO$_2$ and O$_3$ mixing ratios (25%, 9% and 5%, respectively) results in an overall uncertainty of 31% for $k_{nss}^{NO_3}$ which cannot account for its deviation to $k^{NO_3}$.

The fact that $k_{nss}^{NO_3}$ is significantly larger than $k^{NO_3}$ indicates that NO$_3$ can be lost by reactions other than those with reactive,

stable VOCs that can be sampled by the FT-CRDS instrument. As discussed above, RO$_2$ represents the most likely candidate to account for some additional loss of NO$_3$; the numerical simulations (MCM v3.3.1) predict an additional reactivity in the order of ~22% based on a generic value for $k_{NO_3+RO_2}$. However, in order to bring $k^{NO_3}$ and $k_{nss}^{NO_3}$ into agreement, either the RO$_2$ level or the rate coefficient for reaction between NO$_3$ and RO$_2$ (especially NISOPOO) would have to be a factor of 2 larger than incorporated into the model (see below). Alternatively, losses of NO$_3$ (and N$_2$O$_5$) to surfaces enhance $k_{nss}^{NO3}$ but not

$k^{NO_3}$. As no aerosol was present in the experiments analysed above, the only surface available is provided by the chamber walls.

In order to quantify the contribution of NO$_3$ and N$_2$O$_5$ wall losses to $k_{nss}^{NO3}$, we analysed the experiments from the 1$^{st}$ and 2$^{nd}$ August during isoprene-free periods, i.e. when no RO$_2$ radicals are present and (in the absence of photolysis and NO) uptake of NO$_3$ (or N$_2$O$_5$) to the chamber walls represents the only significant sink. Consequently, plotting $k_{nss}^{NO3}$ from this period

against $K_{eq}[NO_2]$ enables separation of direct NO$_3$ losses (R10) from indirect losses via N$_2$O$_5$ uptake (R9) and to derive first-order loss rates $k_{NO_3}^{wall}$ and $k_{N_2O_5}^{wall}$ of NO$_3$ and N$_2$O$_5$ according to Eq. (5). (Allan et al., 2000; Brown et al., 2009b; Crowley et al., 2010; McLaren et al., 2010).

$$k_{nss}^{NO_3} = k_{wall}^{NO_3} + k_{wall}^{N_2O_5} K_{eq}[NO_2] \tag{5}$$

The results from the isoprene-free periods of experiments on the 1$^{st}$ and 2$^{nd}$ of August are shown in Fig. 8. A linear regression

of the data yields a slope $k_{wall}^{N_2O_5}$ of (3.28 ± 1.15) x 10$^{-4}$ s$^{-1}$ and an intercept $k_{wall}^{NO_3}$ of (0.0016 ± 0.0001) s$^{-1}$, indicating that NO$_3$ losses dominate and that heterogeneous removal of N$_2$O$_5$ does not contribute significantly to the overall loss rate constant of ~ 0.002 s$^{-1}$. The data reproducibility from one experiment to the next indicates that the NO$_3$/N$_2$O$_5$ wall loss rates are unchanged if the experimental conditions, i.e. dry air and no aerosols, are comparable. Humidification of the air on the other hand may facilitate heterogeneous reactions of NO$_3$ or N$_2$O$_5$ with the chamber walls and increase corresponding loss rates. This might

be an explanation for observation of a larger difference between $k^{NO_3}$ and $k_{nss}^{NO_3}$ during an experiment under humid conditions on the 6$^{th}$ August (Fig. 7b, blue triangles). Lack of extensive isoprene-free periods on this day impede the extraction of wall loss rates with this approach: Even after subtraction of $k^{NO_3}$ from $k_{nss}^{NO_3}$ equation (5) is not applicable in experiments once isoprene is present (and becomes the dominant sink of NO$_3$) as reactions of RO$_2$ indirectly co-determine the NO$_2$ mixing ratios. For further analysis, the wall loss rate constants of NO$_3$ and N$_2$O$_5$ were fixed as long as there was neither humidity nor particles

in the chamber and are considered invariant with time after isoprene injections. This implicitly assumes that low volatility oxidation products that deposit on chamber walls do not enhance the reactivity of the walls to NO$_3$. As these products have less double bonds than isoprene and react only very slowly with NO$_3$, this assumption would appear reasonable.





We examined the effect of introducing the $NO_3$ and $N_2O_5$ wall loss rate constants calculated as described above into the chemical scheme used in the box model (Model 1, MCM v3.3.1). The results from Model 1 for the experiment on the $2^{nd}$

August are summarised in Fig. 9 (red lines) which compares simulated and measured mixing ratios of $NO_3$, $N_2O_5$, $NO_2$, $O_3$ and isoprene (following its addition at 10:50) as well as the measured and non-steady-state $NO_3$ reactivities $k^{NO_3}$ and $k_{nss}^{NO3}$. The $NO_2$ and $O_3$ mixing ratios are accurately simulated. Furthermore, $NO_3$ and $N_2O_5$ mixing ratios that are only 10 to 30% higher than those measured and therefore $NO_3$ reactivities lower than $k_{nss}^{NO3}$ (orange circles) are predicted. We note that, in these isoprene-free phases, the omission of wall losses results in model predictions of $NO_3$ and $N_2O_5$ mixing ratios up to 1400

and 1600 pptv, which exceed measurements by factors of 4-8, as illustrated in (Fig. S3).

The evolution of the isoprene mixing ratio is reproduced by the model which is why $k^{NO_3}$, (mostly determined by $k_5$[isoprene], (purple area)), is only slightly lower than the simulated overall reactivity by Model 1. After quantification of $NO_3/N_2O_5$ wall losses, $NO_3+RO_2$ reactions remain the only source of additional $NO_3$ reactivity to explain the difference between $k^{NO_3}$ and $k_{nss}^{NO3}$. As already mentioned above, the model may underestimate the effect of $RO_2$ induced losses of $NO_3$ either because the

$RO_2$ mixing ratios are underestimated or because the rate coefficient $k_{RO_2+NO_3}$ is larger than assumed.

The result of a simulation (Model 2) with $k_{RO_2+NO_3}$ set to 4.6 x $10^{-12}$ cm³molecule$^{-1}$s$^{-1}$ (twice the generic value in MCM v3.3.1) is displayed as the blue lines in Fig. 9. The $O_3$, $NO_2$, $N_2O_5$ and isoprene mixing ratios are only slightly affected by this change in the reaction constant, whereas its impact on the $NO_3$ mixing ratios as well as on the reactivity is very significant. The higher rate coefficient for reaction of $NO_3$ with $RO_2$ would not only explain the observed discrepancy between the overall

reactivity $k_{nss}^{NO3}$ and $k^{NO_3}$ but also results in a better reproduction of the $NO_3$ measurement during the isoprene-dominated period. A similar result is obtained in a comparable experiment under dry conditions on the $10^{th}$ August (see Fig. S4 in the supplement).

There are only few experimental studies on reactions of $NO_3$ with $RO_2$ and the rate coefficient for reaction of $NO_3$ with isoprene-derived $RO_2$ has never been measured. For the reaction between $NO_3$ and the methyl peroxy radical ($CH_3O_2$) values

between 1.0 and $2.3 \times 10^{-12}$ cm$^3$ molecule$^{-1}$ s$^{-1}$ have been reported (Crowley et al., 1990; Biggs et al., 1994; Daele et al., 1995; Helleis et al., 1996; Vaughan et al., 2006), with a preferred value of $1.2 \times 10^{-12}$ cm$^3$ molecule$^{-1}$ s$^{-1}$ (Atkinson et al., 2006). Increasing the length of the C-C backbone in the peroxy radical appears to increase the rate coefficient, with values of $2.3 \times 10^{-12}$ cm$^3$ molecule$^{-1}$ s$^{-1}$ preferred for reaction of $NO_3$ with $C_2H_5O_2$ (Atkinson et al., 2006), whereas the presence of electron-withdrawing groups attached to the peroxy-carbon atom reduces the rate coefficient (Vaughan et al., 2006). A single study of

the reaction between $NO_3$ and an acylperoxy radical indicates that the rate coefficient ($4.0 \times 10^{-12}$ cm$^3$ molecule$^{-1}$ s$^{-1}$) may be larger than the MCM adopted value of $2.3 \times 10^{-12}$ cm$^3$ molecule$^{-1}$ s$^{-1}$ (Canosa-Mas et al., 1996). Similarly, an indirect study (Hjorth et al., 1990) of the rate coefficient for the reaction between $NO_3$ and a nitro-substituted, C-6 peroxy radical, $(CH_3)_2C(ONO_2)C(CH_3)_2O_2$, reports a value of $5 \times 10^{-12}$ cm$^3$ molecule$^{-1}$ s$^{-1}$ which may be appropriate for longer-chain peroxy radicals derived from biogenic trace gases. In light of the large uncertainty associated with the kinetics of $RO_2 + NO_3$ reactions,

a rate coefficient of $4.6 \times 10^{-12}$ cm$^3$ molecule$^{-1}$ s$^{-1}$ for reaction between NISOPOO and $NO_3$ is certainly plausible.





We note however, that use of a faster rate coefficient for the reaction between RO$_2$ and NISOPOO, RO$_2$ isomerisation processes and differentiation between the fates of the main NISOPOO isomers as proposed by Schwantes et al. (2015) would result in lower RO$_2$ mixing ratios. If $k_{NISOPOO+RO_2}$ in MCM v3.3.1 is set to a value of $5 \times 10^{-12}$ cm$^3$ molecule$^{-1}$ s$^{-1}$ (average over all isomers, Schwantes et al., 2015) a slightly higher value of 5.2 x $10^{-12}$ cm$^3$ molecule$^{-1}$ s$^{-1}$ for $k_{RO_2+NO_3}$ would be necessary to

bring modelled and measured NO$_3$ reactivity into agreement. Conversely, increasing RO$_2$ concentrations by the required factor two would necessitate a significant reduction in the model rate coefficients for RO$_2$ + RO$_2$ or RO$_2$ + HO$_2$ reactions, which contradicts experimental results (Boyd et al., 2003; Schwantes et al., 2015) and is considered unlikely.

Differences in measurement of $k_{nss}^{NO_3}$ and modelled NO$_3$ reactivity could also result from incorrectly modelled product yields owing to the simplified mechanism used, which does not consider in detail e.g. the formation of methyl vinyl ketone (MVK)

via β-NISOPOO isomers or the reaction between NO$_3$ and other main products like hydroxy isopropyl nitrates (e.g. O$_2$NOCH$_2$C(CH$_3$)CHCH$_2$OH, ISOPCNO3) and nitrooxy isopropyl hydroperoxide (O$_2$NOCH$_2$C(CH$_3$)CHCH$_2$OOH, NISOPOOH). However, none of these products is expected to react sufficiently rapidly with NO$_3$ to make a difference: The rate coefficient for reaction of NO$_3$ with MVK is < 6 x $10^{-16}$ cm$^3$ molecule$^{-1}$ s$^{-1}$ and that for 2-methyl-3-butene-2-ol (a comparable molecule to ISOPCNO3) is 1.2 x $10^{-14}$ cm$^3$ molecule$^{-1}$ s$^{-1}$ at 298 K (IUPAC 2019). Even ppbv amounts of these

products would not cause significant additional NO$_3$ reactivity.

On the other hand, the FT-CRDS will underestimate the reactivity of NO$_3$ if products that are formed do not make it to the inlet (i.e. traces gases with high affinity for surfaces). One potential candidate for this category is NISOPOOH, formed in the reaction between NISOPOO and HO$_2$. There are no kinetic data on the reaction of NO$_3$ with NISOPOOH, though, given the lack of reactivity of NO$_3$ towards organic peroxides it is very unlikely that the rate coefficient would be larger than for NO$_3$ +

O$_2$NOCH$_2$C(CH$_3$)=CHCHO. Analysis of one experiment (9$^{th}$ of August, Fig. 7b), in which HO$_2$ production (and thus the yield of NISOPOOH) was enhanced by the addition of propene and CO, shows that the difference between $k^{NO_3}$ and $k_{nss}^{NO_3}$ on that day is comparable to those of the other experiments. This would also indicate that the influence of the potential non-detection of the hydroperoxide on the analysis should be low.

All in all, the results of the analysis above strongly suggest that the difference between directly measured and non-steady-state

reactivity $k_{nss}^{NO_3}$ is caused by reactions of NO$_3$ with RO$_2$ with the results best explained when a rate coefficient of ~5 $\times 10^{-12}$ cm$^3$ molecule$^{-1}$ s$^{-1}$ is used. Quantifying the impact of peroxy radicals on the fate of NO$_3$ is however challenging as not only the rate coefficients for RO$_2$ + NO$_3$ are scarce and uncertain but also the rate constants for self-reaction of RO$_2$ derived from NO$_3$ + isoprene have not been determined in direct kinetic measurement but via analyses of non-radical product yields.

## 4 Summary and conclusion

Direct measurements of NO$_3$-reactivity ($k^{NO_3}$) in chamber experiments exploring the NO$_3$ induced oxidation of isoprene showed excellent agreement with NO$_3$ loss rate constants calculated from isoprene mixing ratios, thus underlining the



reliability of the reactivity measurements even under unfavourable conditions with as much as 25 ppbv of $NO_2$ in the chamber. The main contributor to the overall uncertainty in $k^{NO_3}$ is the correction (via numerical simulation) for the reaction of $NO_3$ with $NO_2$ and the thermal decomposition of the $N_2O_5$ product. The results of the NO3ISOP campaign indicate that previously

derived overall uncertainties (Liebmann et al., 2017) that considered an uncertainty of 10% in the rate coefficients of both reactions (Burkholder et al., 2015) and an 8% uncertainty for the $NO_2$ mixing ratios are too large.

The measured reactivity $k^{NO_3}$ could be completely assigned to the reaction between $NO_3$ and isoprene, indicating that contributions from reactions of non-radical oxidation products are minor, which is consistent with predictions of the current version of the Master Chemical Mechanism.

Values of $NO_3$ reactivity as calculated from $NO_3$ and $N_2O_5$ mixing ratios and the $NO_3$ production term were found to be a factor of ~1.85 higher than the directly measured $NO_3$ reactivities $k^{NO_3}$. A box-model analysis indicates that the most likely explanation is a larger fractional loss of $NO_3$ via reactions with organic peroxy radicals ($RO_2$) formed during the oxidation of isoprene. A rate coefficient $k_{RO_2+NO_3} = 4.6 \times 10^{-12}$ cm$^3$ molecule$^{-1}$ s$^{-1}$ is necessary to align model predictions (MCM v.3.3.1) and observations.

**Acknowledgements**

This work was supported by the EC Horizon 2020 project Eurochamp2020 (grant agreement no. 730997) and Labex Voltaire (ANR-10-LABX-100-01). This project has received funding from the European Research Council (ERC) under the European Union's Horizon 2020 research and innovation programme (SARLEP grant agreement No. 681529). We thank Chemours for provision of the FEP sample used to coat the CRD cavities and flowtube reactor.

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



**Tables and Figures**

**Table 1. Experimental conditions in the SAPHIR chamber during the NO3ISOP campaign.**

| Date | T ( °C) | H$_2$O (%) | D/N | O$_3$ (ppbv) | NO$_2$ (ppbv) | Isoprene (ppbv) | Seed aerosol | Notes |
|---|---|---|---|---|---|---|---|---|
| 31 July | 25-35 | 0 | N | 90-120 | 1-5 | 0 | -- | |
| 1 August | 22-31 | 0 | N | 85-115 | 2-5 | 1.5 | -- | |
| 2 August | 23-38 | 0 | N | 85-120 | 2-5 | 3 | -- | |
| 3 August | 30-42 | 1.3-2.7 | D->N | 45-100 | 1-5 | 3 | -- | |
| 6 August | 20-44 | 1.4 | N->D | 40-110 | 1-6 | 3.5 | -- | |
| 7 August | 20-41 | 0.45-0.6 | N | 45-60 | 3-4.5 | 2 | -- | contamination |
| 8 August | 22-28 | 0 | N | 75-115 | 13-30 | 8 | -- | |
| 9 August | 20-27 | 0 | N | 65-115 | 6-2.5 | 3 | -- | CO & propene |
| 10 August | 17-28 | 0 | N | 40-65 | 3-5.5 | 2.3 | -- | |
| 12 August | 14-36 | 0 | N->D | 70-115 | 4-12 | 3 | -- | CO |
| 13 August | 28-24 | 0 | N | 75-110 | 12-23 | 8 | -- | |
| 14 August | 18-24 | 0 | N | 70-110 | 13-22 | 11 | (NH$_4$)$_2$SO$_4$ | Reduced fan operation |
| 15 August | 20-28 | 1.3-2 | N | 80-115 | 8-21 | 7 | (NH$_4$)$_2$SO$_4$ | |
| 16 August | 20-28 | 1.6 | N->D | 80-115 | 2-5 | 2.5 | (NH$_4$)$_2$SO$_4$ | |
| 17 August | 18-26 | 1.2-1.7 | N->D | 0-400 | 0-17 | 2.5 | -- | Isobutyl nitrate, calibration |
| 18 August | 14-31 | 1.3-1.4 | N->D | 80-110 | 2-5 | 2.5 | (NH$_4$)$_2$SO$_4$ | β-caryophyllene |
| 19 August | 16-31 | 0.07 | N | 0-110 | 0-20 | 2.3 | (NH$_4$)$_2$SO$_4$ | MVK, N$_2$O$_5$ as NO$_2$ source |
| 20 August | 20-26 | 1.2-19 | N | 85-130 | 3-5 | 4.5 | (NH$_4$)$_2$SO$_4$ | β-caryophyllene |
| 21 August | 20-30 | 1.5-1.9 | N | 55-130 | 2-5 | 4.5 | (NH$_4$)$_2$SO$_4$ | CO & propene |
| 22 August | 18-33 | 1.3-17 | N | 75-110 | 2.5-8.5 | 4 | (NH$_4$)$_2$SO$_4$ | plant emissions |
| 23 August | 18-31 | 1.5-2.2 | N | 45-100 | 3.5-5 | 3 | (NH$_4$)$_2$SO$_4$ | |
| 24 August | 17-23 | 1-1.6 | N | 85-110 | 2.3-5.5 | 22 | NH$_4$HSO$_4$ | β-caryophyllene |

D/N denotes if the experiment was conducted with the chamber roof opened (D: daytime) or closed (N: nighttime) and in which order a transition was done. Only maximum values of measured isoprene are listed.



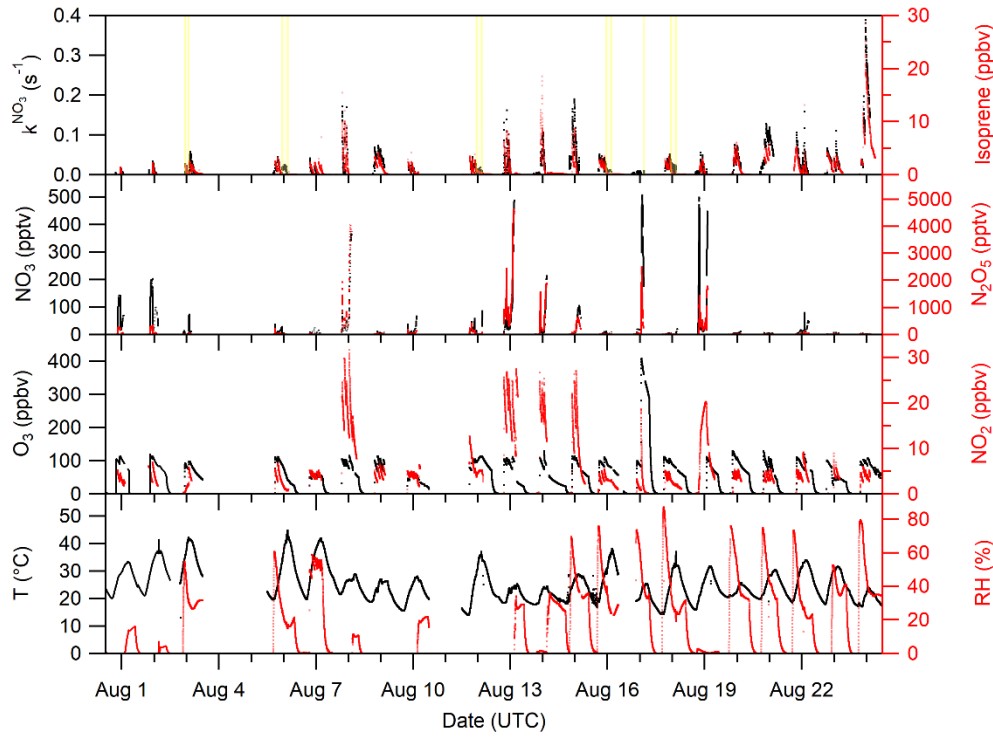

**Figure 1: Overview of the temperature (T), relative humidity (RH), VOC-induced NO₃ reactivity ($k^{NO_3}$) as well as the O₃, NO₂, NO₃, N₂O₅ and isoprene mixing ratios during the NO3ISOP campaign. The yellow shaded area in the upper panel represent phases of the experiment when the chamber roof was opened. The ticks mark 12:00 UTC of the corresponding day.**


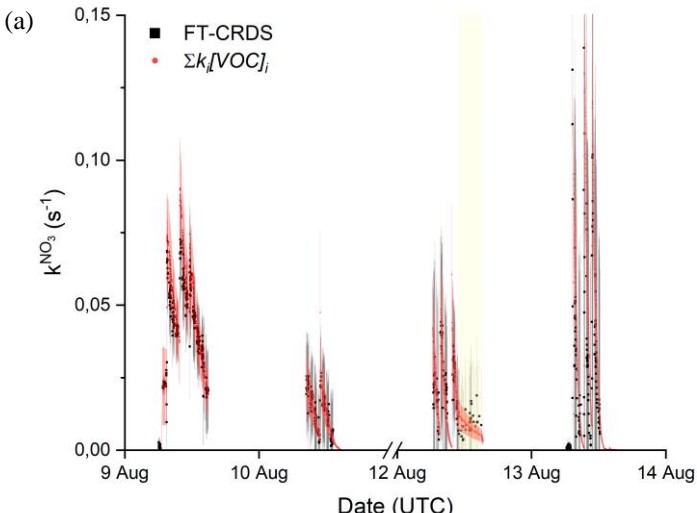


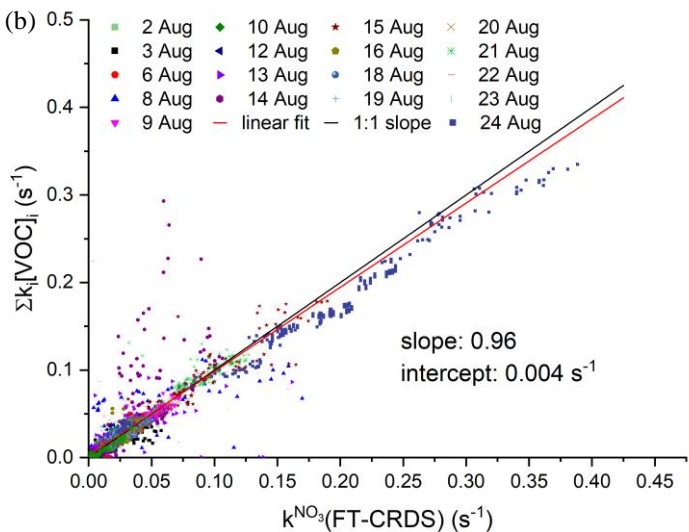

**Figure 2: (a) 4-day time-series of $k^{NO_3}$ and $\Sigma k_i[VOC]_i$. The total uncertainty in $k^{NO_3}$ was calculated as described by Liebmann et al. (2017) and is indicated by the grey shaded area. The red shaded area shows the associated uncertainty of the calculated reactivities and are derived from error propagation using the standard deviation of the isoprene mixing ratios and an uncertainty of 41 % for the rate coefficient for reaction between $NO_3$ and isoprene (IUPAC, 2019). The ticks mark 00:00 UTC of the corresponding date and yellow-shaded areas represent periods in which the chamber roof was opened. (b) Correlation between $\Sigma k_i[VOC]_i$ and $k^{NO_3}$ measurements. The red line represents a least-squares, linear fit to the entire data set, while the black line illustrates an ideal slope of 1:1.**



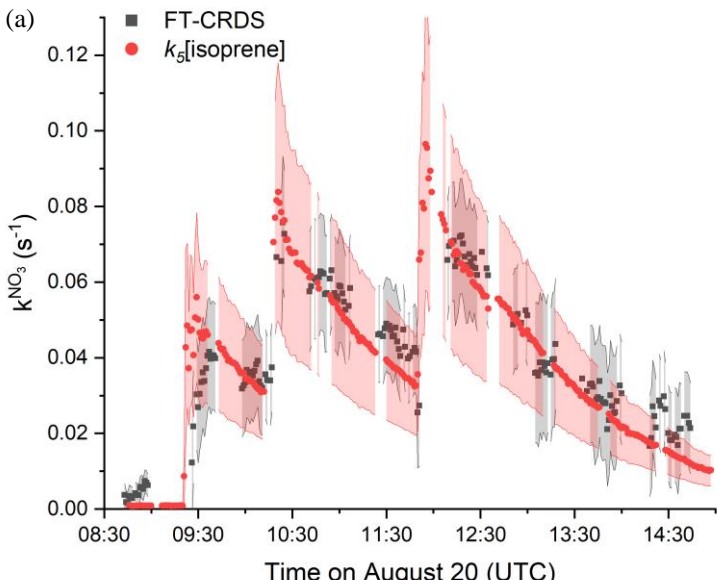


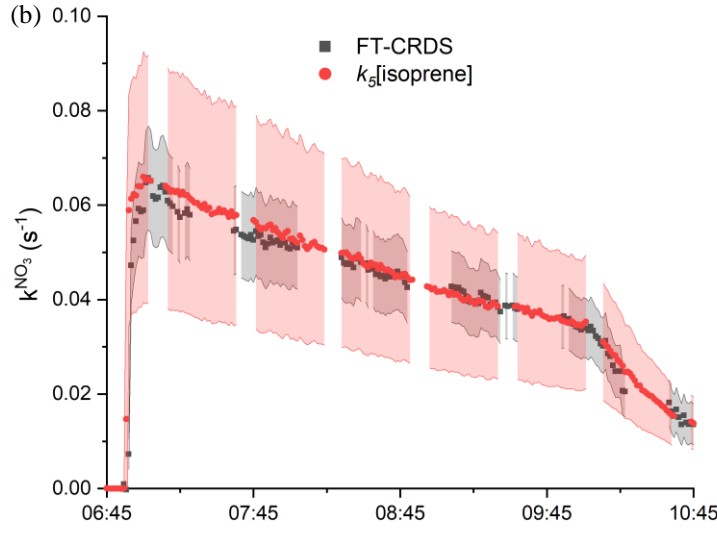

**Figure 3: Measured reactivity ($k^{NO_3}$, black data points) and reactivity calculated from Eq. (1) (red data points) which is equivalent to $k_5$[isoprene]. The grey shaded area represents the total uncertainty in $k^{NO_3}$; the red-shaded areas the total uncertainty in $k_5$[isoprene] and were estimated as explained in Fig.2. (a) 20th August: Type 1 experiment with initial mixing ratios of NO₂ = 4.6 ppbv and O₃ = 120 ppbv. (b) 23rd August: Only O₃ (100 ppbv) and isoprene (4 ppbv) were initially present.**


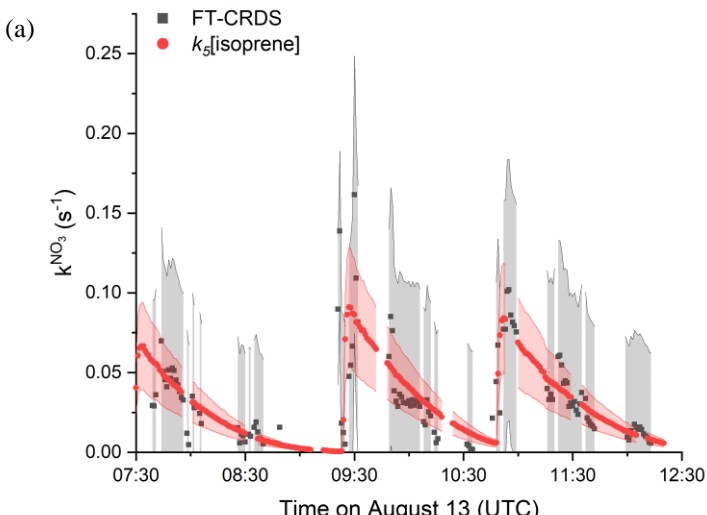

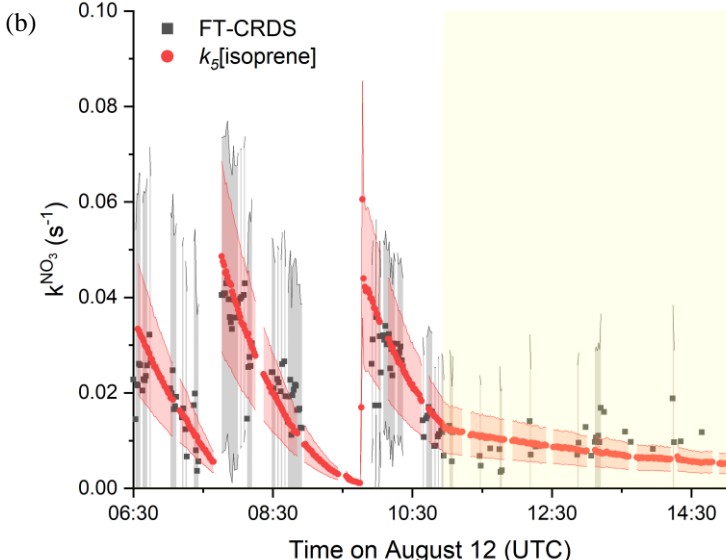

**Figure 4: Measured (black) and expected (red) NO₃-reactivity using Eq.(1). The corresponding uncertainties were estimated as described in Fig.2 and are indicated as shaded areas. (a) Type 2 experiment from the 13th August under dry conditions with initial mixing ratios of NO₂ = 25 ppbv and O₃ = 104 ppbv. (b) Experiment from the 12th August with NO₂ mixing ratios between 7 and 12 ppbv and initial mixing ratio of O₃ = 79 ppbv. The yellow shaded area denotes the period with the chamber roof opened after 11:00 UTC.**

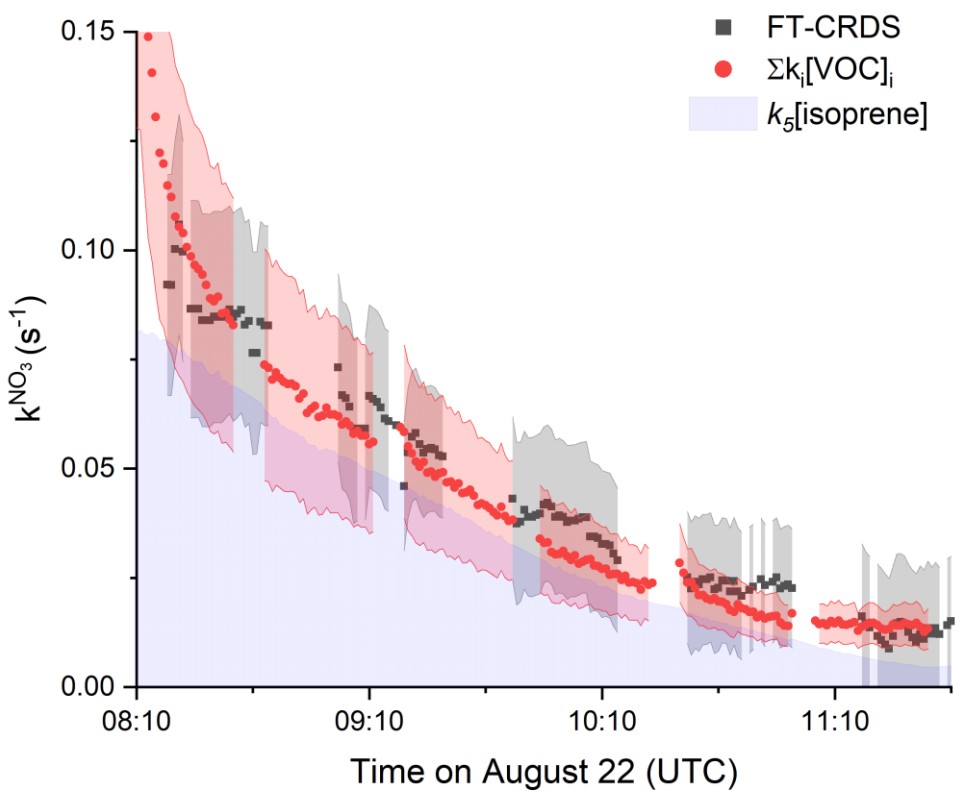


**Figure 5: Results from 22nd August between 08:00 and 11:40 UTC. Comparison between $k^{NO_3}$ (black data points, uncertainty as grey shaded area) and NO₃ reactivity calculated from $\Sigma k_i[\text{VOC}]_i$ (red data points) using the measured isoprene and Σmonoterpenes mixing ratios. The associated uncertainty (red area) was derived by error propagation considering the standard deviations of the VOC mixing ratios as well as the uncertainties of the rate coefficients (41% for $k_5$ and 47% for $k_{monoterpenes}$). The uncertainty of**
**$k^{NO_3}$ was estimated as explained in Fig.2. The contribution of isoprene to the observed reactivity is indicated by the area in purple.**




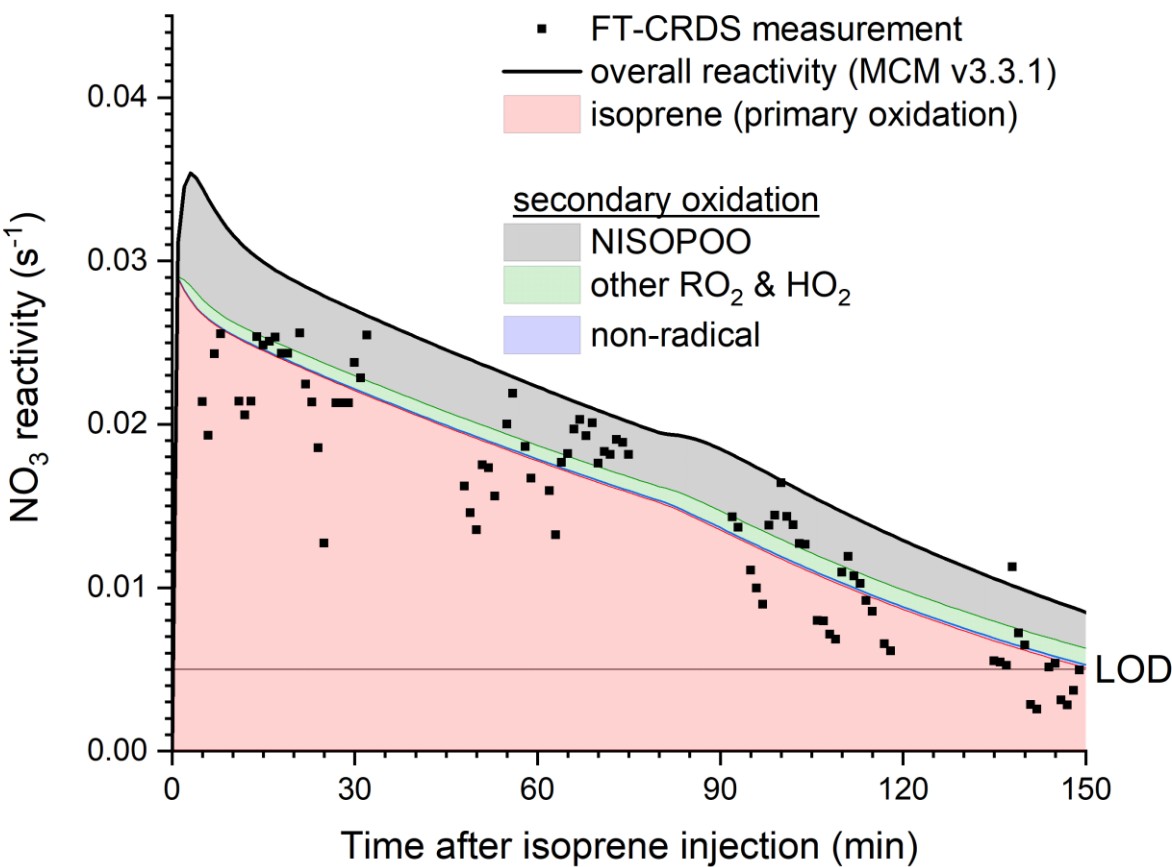

**Figure 6: Experimental results for $k^{NO_3}$ and numerical simulation (MCM v3.3.1) of the NO$_3$ reactivity following the first isoprene injection of the experiment on the 10th August. The simulation was run with 1 min resolution, initial conditions were 60 ppbv of O$_3$, 5.5 ppbv of NO$_2$ and 2 ppbv of isoprene and used actual chamber temperatures, which increased from 293 to 301 K during the course of the experiment. Wall losses of NO$_3$ and N$_2$O$_5$ were parameterised as described in the text. Individual contributions to the NO$_3$ reactivity of isoprene, peroxy radicals and secondary oxidation products are highlighted.**



(a)

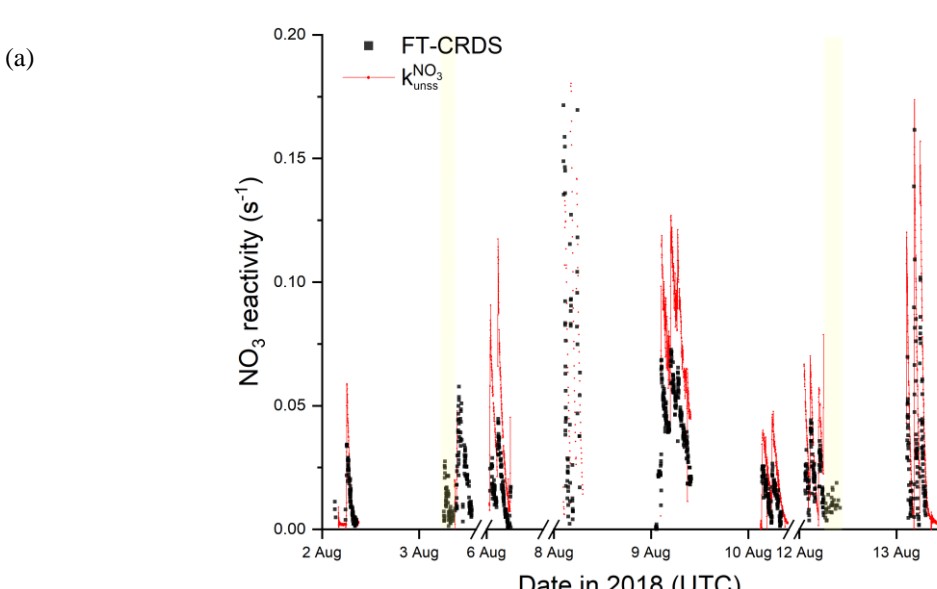

(b)

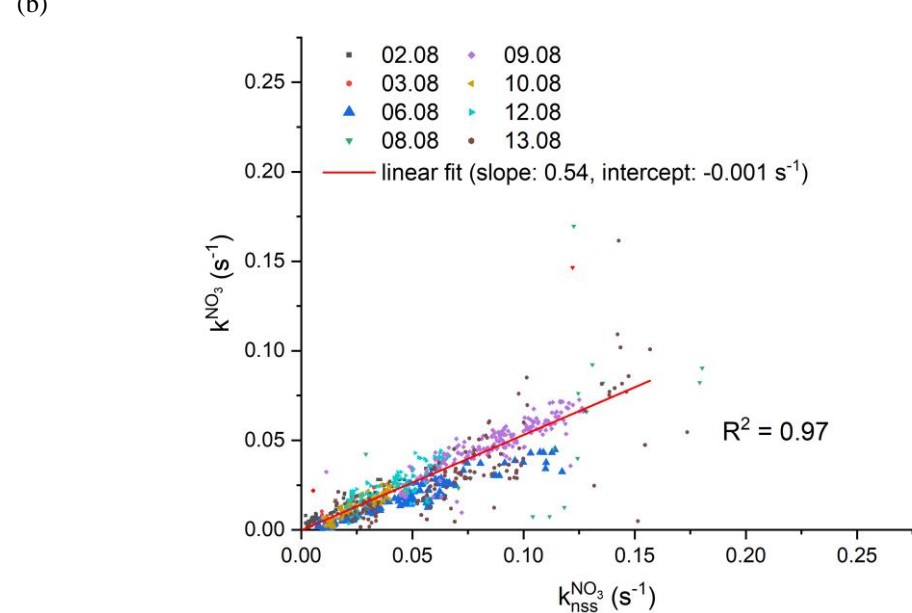

**Figure 7: (a) Overview of measured (black) and calculated NO₃-reactivity with Eq. 3 (red). The ticks mark 00:00 UTC of the**
**corresponding day. The yellow-coloured areas denote periods with an opened chamber roof. For the sake of clarity, the uncertainties are not included. (b) Correlation plot between $k^{NO_3}$ and $k^{NO_3}_{nss}$. The red line represents an unweighted, orthogonal linear regression ($R^2$ = 0.97) of the complete dataset.**

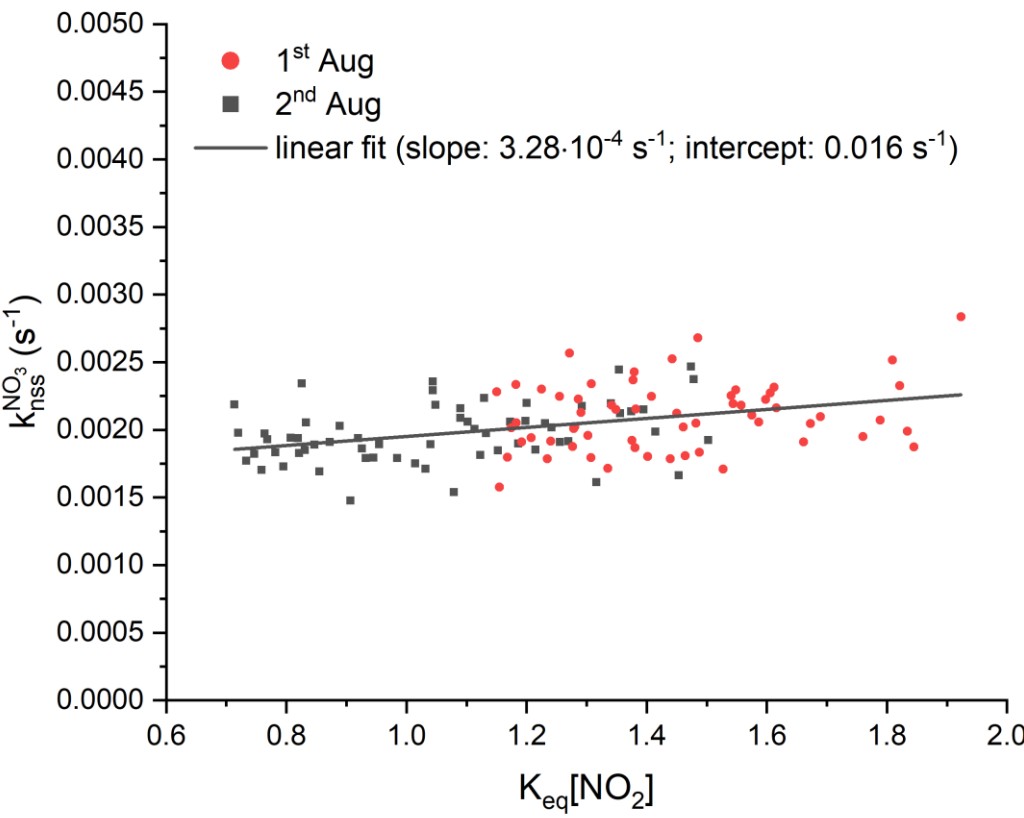

**Figure 8: Analysis of the contribution of wall losses of NO₃ and N₂O₅ to NO₃ reactivity $k_{nss}^{NO_3}$ using experimental data during isoprene-free periods on the 1st (red) and 2nd (black) August. Least-squares, linear fit of the data is shown with a black line and yielded to an intercept $k_{wall}^{NO_3}$ of 0.016 s⁻¹ as well as to a slope $k_{wall}^{N_2O_5}$ of 3.28 x 10⁻⁴ s⁻¹. For sake of better clarity, error bars are not included.**





**Figure 9: O₃, NO₂, NO₃, N₂O₅ and isoprene mixing ratios and NO₃ reactivity on 2ⁿᵈ August (black). The grey shaded area symbolizes**
**the overall uncertainty associated with each measurement. Orange circles denote the reactivity obtained using Eq.(3). The results of**
**the numerical simulation using MCM v.3.3.1 (with NO₃ and N₂O₅ wall loss rates of 0.016 s⁻¹ and 3.3 x 10⁻⁴ s⁻¹ respectively) for each**
**of the reactants is shown by a red line, whereas the blue line shows the result of the same model with the rate coefficient for reaction**
**between NO₃ and RO₂ set to 4.6 × 10⁻¹² cm³molecule⁻¹s⁻¹, which is twice the value estimated by the MCM.**