# Peer review of "Evolution of NO3 reactivity during the oxidation of isoprene"

_Atmospheric Chemistry and Physics, 2020_

## Author Comment (AC1) · 21 May 2020

We draw attention to a potentially confusing statement in our text. On lines 401 to 407 we write:

"The result of a simulation (Model 2) with $k$RO2+NO3 set to 4.6 x $10^{-12}$ cm³ molecule$^{-1}$ s$^{-1}$ (twice the generic value in MCM v3.3.1) is displayed as the blue lines in Fig. 9. The $O_3$, $NO_2$, $N_2O_5$ and isoprene mixing ratios are only slightly affected by this change in the reaction constant, whereas its impact on the $NO_3$ mixing ratios as well as on the reactivity is very significant. The higher rate coefficient for reaction of $NO_3$ with $RO_2$ would not only explain the observed discrepancy between the overall reactivity $k_{nss}^{NO_3}$ and $k^{NO_3}$ but also results in a better reproduction of the $NO_3$ measurement during the isoprene-dominated period. A similar result is obtained in a comparable experiment under dry conditions on the 10th August (see Fig. S4 in the supplement)."

A value of $k(RO_2+NO_3)$ = 4.6 x $10^{-12}$ cm³molecule$^{-1}$ s$^{-1}$ is sufficient to bring the measured and modelled $NO_3$-reactivities into agreement *within the uncertainty associated with the measurements*. The blue line plotted in Figure 9 (which represents optimum agreement irrespective of uncertainties) was however calculated with a value of 9.2 x $10^{-12}$ cm³molecule$^{-1}$ s$^{-1}$.

The caption to Figure 9 and Figure S4 should thus read:

**Figure 9: $O_3$, $NO_2$, $NO_3$, $N_2O_5$ and isoprene mixing ratios and $NO_3$ reactivity on 2nd August (black). The grey shaded area symbolizes the overall uncertainty associated with each measurement. Orange circles denote the reactivity obtained using Eq.(3). The results of the numerical simulation using MCM v.3.3.1 (with $NO_3$ and $N_2O_5$ wall loss rates of 0.016 s$^{-1}$ and 3.3 x $10^{-4}$ s$^{-1}$ respectively) for each of the reactants is shown by a red line, whereas the blue line shows the result of the same model with the rate coefficient for reaction between $NO_3$ and $RO_2$ set to 9.2 × $10^{-12}$ cm³ molecule$^{-1}$ s$^{-1}$. *Considering total uncertainty, a rate coefficient of 4.6 × $10^{-12}$ cm³ molecule$^{-1}$ s$^{-1}$ is sufficient to bring model and measurement into agreement.***

**Figure S4: $O_3$, $NO_2$, $NO_3$, $N_2O_5$ and isoprene mixing ratios as well as the $NO_3$ reactivity on the experiment of the 10$^{th}$ August (black). The grey shaded area symbolizes the overall uncertainty associated with each measurement. Orange circles denote the non-steady-state reactivity obtained from Eq.(3). The results of the numerical simulation using MCM v.3.3.1 (with $NO_3$ and $N_2O_5$ wall loss rate of 0.016 s$^{-1}$ and 3.3 x $10^{-4}$ s$^{-1}$ respectively) for each of the reactants is shown by a red line, whereas the blue line shows the result of the same model with $k_{NO_3+RO_2}$= 9.2 x $10^{-12}$ cm³molecule$^{-1}$s$^{-1}$. *Considering total uncertainty, a rate coefficient of 4.6 × $10^{-12}$ cm³ molecule$^{-1}$ s$^{-1}$ is sufficient to bring model and measurement into agreement.***

The overall conclusions, that reactions of $NO_3$ with $RO_2$ contribute significantly to $NO_3$-reactivity and that the rate coefficient for reaction between $NO_3$ and $RO_2$ is potentially larger than presently used in the MCM, are unchanged.

---

## Referee Comment (RC1) · Anonymous Referee #1 · 8 Jun 2020

**Referee Report:**
**Evolution of $NO_3$ reactivity during the oxidation of isoprene**

Anonymous Referee

**1 Overview**

Dewald et al. present measurement of $NO_3$ reactivity ($k^{NO_3}$) resulting from the reaction of $NO_3$ with isoprene and stable trace gases in an atmospheric simulation chamber with different initial conditions. The agreement between $\sum k_i [VOC]_i$ and $k^{NO_3}$ indicates that $NO_3$ reactivity is dominated by the reaction between $NO_3$ and isoprene. Box model simulation results indicate that the discrepancy between measured $k^{NO_3}$ and non-steady-state reactivity ($k_{nss}^{NO_3}$) is caused by the uncertainty in $k_{RO_2+NO_3}$.

Instrument analysis is adequate. However the authors should expand the description of instrument calibration for PTR-TOF-MS (see minor comments below).

Overall, this study reports high quality data obtained from well designed experiments. The data should be of interest to the atmospheric science community. This manuscript is well within the scope of ACP. I recommend that the manuscript be published in ACP after minor revision.

**2 Minor comments**

(1) 2.3 VOC measurements: PTR-ToF-MS

- Please describe how often were the instruments calibrated during the campaign. Please show the variability of the instrumental sensitivities during the entire campaign period;

- Please be more specific about the uncertainty used in instrument comparison. It would be useful to add a figure showing the VOCs mixing ratios measured by the two PTR-TOF-MS from the same air sample.

(2) 2.5 Box model: "FACSIMILE/CHEKMAT" is a dated tool. A quick search of it didn't return much useful information. It would be great if the simulations in this study were run in an open source, modern box model, such as BOXMOX (Knote et al., 2015), F0AM (Wolfe et al., 2016), and CAABA (Sander et al., 2019). Doing so enables the reader to run the simulation on their own computer and play around with the configurations, such as the reaction rate constant $k_{RO_2+NO_3}$, the wall loss rates of $NO_3$ etc.

(3) Page 8, Line 227: "no propene data was available": is this due to the unavailability of propene in the standard gas? If so, the expected sensitivity of propene can be calculated using the method described in Holzinger et al. (2019). The

uncertainty of propene mixing ratios introduced from using expected sensitivity should be smaller than using model estimation. Please justify why the propene mixing ratios were assessed with the model instead of calculated using its sensitivity.

(4) Page 8, Line 242: Please provide output from the unweighted linear regression (e.g., correlation coefficient, p-value), and incorporate the output into your discussion on the agreement between $\sum k_i[VOC]_i$ and $k^{NO_3}$ measurements.

(5) Page 13, Line 388–395:

  – Please merge the model output (with $k_{wall} = 0 \text{ s}^{-1}$) in Figure S3 to Figure 9, this could help the reader better visualize the effect of introducing the $NO_3$ and $N_2O_5$ wall loss;

  – Please discuss more about the effect of omitting the $NO_3$ and $N_2O_5$ wall loss and its cause of large discrepancies between the measurement and model simulation in $NO_3$, $N_2O_5$, and isoprene mixing ratios;

  – Please discuss how is the first-order wall loss rate for $O_3$, $H_2O_2$, HO, HONO and $HNO_3$ derived in Table S1.

(6) Page 13, Line 391: "and isoprene (following its addition at 10:50)": from Figure 9 and Figure S2, $NO_2$ appeared to be injected at 10:50, isoprene appeared to be injected at 11:00, please clarify.

(7) Page 22, Figure 2(b): To better aid visual inspection of the dataset, please set the aspect ratio of x:y to 1:1, add grid to x-axis and y-axis, add border to the legend (not shown in the demo below). See Figure 2(b).

[Figure]

**Figure 2(b).** Correlation between $\sum k_i[VOC]_i$ and $k^{NO_3}$ measurements. For demo purpose only, dataset are not categorised according to sampling days, no legend is shown, no linear regression result is shown.

**References**

Holzinger, R., Acton, W. J. F., Bloss, W. J., Breitenlechner, M., Crilley, L. R., Dusanter, S., Gonin, M., Gros, V., Keutsch, F. N., Kiendler-Scharr, A., Kramer, L. J., Krechmer, J. E., Languille, B., Locoge, N., Lopez-Hilfiker, F., Materić, D., Moreno, S., Nemitz, E., Quéléver, L. L. J., Sarda Esteve, R., Sauvage, S., Schallhart, S., Sommariva, R., Tillmann, R., Wedel, S., Worton, D. R., Xu, K., and Zaytsev, A.: Validity and limitations of simple reaction kinetics to calculate concentrations of organic compounds from ion counts in PTR-MS, Atmospheric Measurement Techniques, 12, 6193–6208, https://doi.org/10.5194/amt-12-6193-2019, https://www.atmos-meas-tech.net/12/6193/2019/, 2019.

Knote, C., Tuccella, P., Curci, G., Emmons, L., Orlando, J. J., Madronich, S., Baró, R., Jiménez-Guerrero, P., Luecken, D., Hogrefe, C., Forkel, R., Werhahn, J., Hirtl, M., Pérez, J. L., Roberto, Giordano, L., Brunner, D., Yahya, K., and Zhang, Y.: Influence of the choice of gas-phase mechanism on predictions of key gaseous pollutants during the AQMEII phase-2 intercomparison, Atmospheric Environment, 115, 553–568, https://doi.org/https://doi.org/10.1016/j.atmosenv.2014.11.066, http://www.sciencedirect.com/science/article/pii/S1352231014009388, 2015.

Sander, R., Baumgaertner, A., Cabrera-Perez, D., Frank, F., Gromov, S., Grooß, J.-U., Harder, H., Huijnen, V., Jöckel, P., Karydis, V. A., Niemeyer, K. E., Pozzer, A., Riede, H., Schultz, M. G., Taraborrelli, D., and Tauer, S.: The community atmospheric chemistry box model CAABA/MECCA-4.0, Geoscientific Model Development, 12, 1365–1385, https://doi.org/10.5194/gmd-12-1365-2019, https://www.geosci-model-dev.net/12/1365/2019/, 2019.

Wolfe, G. M., Marvin, M. R., Roberts, S. J., Travis, K. R., and Liao, J.: The Framework for 0-D Atmospheric Modeling (F0AM) v3.1, Geoscientific Model Development, 9, 3309–3319, https://doi.org/10.5194/gmd-9-3309-2016, https://www.geosci-model-dev.net/9/3309/2016/, 2016.

---

## Referee Comment (RC2) · Anonymous Referee #2 · 9 Jun 2020

The authors report on studies of NO3 reactivity during 'nighttime' experiments in the SAPHIR chamber, with a primary focus on isoprene chemistry. An FT-CRDS system is used to determine the NO3 reactivity with respect to stable products in the chamber, while a box model analysis is used to assess additional NO3 losses (reaction with peroxy radicals, chamber wall losses) not determined by the FT-CRDS system. Among the key findings are the following: the FT-CRDS accurately measures the NO3 reactivity towards isoprene, and functions well under the conditions studied; stable products of the NO3/isoprene chemistry do not contribute significantly to NO3 reactivity; the generic (and highly uncertain) RO2 + NO3 rate coefficient may be a factor of two or more higher than current estimates. Overall, this is a very solid paper that certainly is publishable in ACP. The paper is well written, and assumptions and uncertainties in the

measurements are generally presented in detail. A few questions and suggestions are presented below for the authors to consider.

There are assumptions and caveats associated with equation (1), line 220 – Could there be significant reaction products that the PTR-MS is unable to detect? Could some products not make it into the flow tube for detection by the k(NO3) instrument? NO3 losses due to chamber walls and radicals are not measured by the k(NO3) instrument. Most (or maybe all) of these are dealt with at different points in the manuscript, but a clear statement or two delineating these at this point might be helpful to the reader.

Can the authors be more quantitative regarding the b-caryophyllene expt (Fig 3a)? - e.g., What is its expected lifetime? The k(NO3) instrument is clearly not seeing the full impact of the stated addition of 2 ppbv b-caryophyllene.

Line 265 / Fig 3b: Isoprene loss here is due to reaction with O3, I assume (maybe also OH formed in the ozonolysis)? Does the agreement noted between the k(NO3) instrument and the k[isoprene] calculation imply that major isoprene ozonolysis products are also comparatively unreactive towards NO3? (Also, a minor detail, but the isoprene decay seems more rapid than would be implied by the O3 concentration given?)

Line 312 or so - It should be noted here that NC4CHO is only one of many products that can be formed.

In Figure 9, it is not clear to me that the increased RO2 + NO3 rate coefficient improves the model/measured NO3 comparison?

---

## Editor Comment (EC1) · Thomas Karl (Editor) · 31 Jul 2020

For clarificaton: While it is correct that m/z 43+ can be a common fragmentation mass during PTR, the fragment C3H6H+ (nominal protonated mass: 43+) is typically not associated with isoprene, which shows a minor fragmentation channel on nominal mass 41+ (i.e. m/z 69 -> m/z 41: d(amu) = 28). A similar (higher yield) fragmentation pattern during proton-transfer is also produced by monoterpenes (e.g. m/z 137 -> m/z 81 = d(amu) = 56 = 2*28).

---

## Author Response (AR1)

**Reply to Anonymous Referee #1**

*In the following, the referee's comments are reproduced (black) along with our replies (*blue*) and changes made to the text (*red*) in the revised manuscript.*

Dewald et al. present measurement of $NO_3$ reactivity ($k^{NO_3}$) resulting from the reaction of $NO_3$ with isoprene and stable trace gases in an atmospheric simulation chamber with different initial conditions. The agreement between $\sum k_i [VOC]_i$ and $k^{NO_3}$ indicates that $NO_3$ reactivity is dominated by the reaction between $NO_3$ and isoprene. Box model simulation results indicate that the discrepancy between measured $k^{NO_3}$ and non-steady-state reactivity $k_{nss}^{NO_3}$) is caused by the uncertainty in $k_{RO_2+NO_3}$.

Instrument analysis is adequate. However the authors should expand the description of instrument calibration for PTR-TOF-MS (see minor comments below).

Overall, this study reports high quality data obtained from well designed experiments. The data should be of interest to the atmospheric science community. This manuscript is well within the scope of ACP. I recommend that the manuscript be published in ACP after minor revision.

We thank the referee for the positive evaluation of our manuscript and the useful comments.

**1. Minor comments**

2.3 VOC measurements: PTR-ToF-MS: Please describe how often were the instruments calibrated during the campaign.

Calibration of PTR1000 was done once per day (around 5 p.m.) following the procedure as described in Holzinger et al. (2019) and took around 10 min. VOCUS PTR performed calibrations on an hourly basis for 5 minutes. This information has been integrated into the manuscript:

L151: Data processing was done using PTRwid (Holzinger, 2015) and the quantification/calibration was done once per day following the procedure as described recently (Holzinger et al., 2019).

L154: Calibration was performed on an hourly basis for 5 minutes.

Please show the variability of the instrumental sensitivities during the entire campaign period.

The sensitivity mostly varies with the primary ion signal as long as other conditions are kept constant (not the case for the whole campaign). The authors therefore do not see the benefit of providing this information in scope of this analysis.

Please be more specific about the uncertainty used in instrument comparison. It would be useful to add a figure showing the VOCs mixing ratios measured by the two PTR-TOF-MS from the same air sample.

The uncertainty associated with the isoprene measurement is 14 %. A new figure (S1) showing the isoprene mixing ratios measured by the two PTR-ToF-MS during two exemplary experiments has been added to the supplement and is mentioned in the manuscript (L155):

The isoprene measurements of the two instruments agreed mostly within the uncertainties (14 %). An exemplary comparison between the two instruments of an isoprene measurement can be found in the supplement (Fig. S1).
* * *
2.5 Box model: "FACSIMILE/CHEKMAT" is a dated tool. A quick search of it didn't return much useful information. It would be great if the simulations in this study were run in an open source, modern box model, such as BOXMOX (Knote et al., 2015), F0AM (Wolfe et al., 2016), and CAABA (Sander et al.,

2019). Doing so enables the reader to run the simulation on their own computer and play around with the configurations, such as the reaction rate constant $k_{RO_2+NO_3}$, the wall loss rates of NO₃ etc.

We present the full chemical scheme used in the simulations. Anyone who wants to reproduce or check our simulations has all the necessary information and can make their own choice of numerical integration tool.
* * *
Page 8, Line 227: "no propene data was available": is this due to the unavailability of propene in the standard gas? If so, the expected sensitivity of propene can be calculated using the method described in Holzinger et al. (2019). The uncertainty of propene mixing ratios introduced from using expected sensitivity should be smaller than using model estimation. Please justify why the propene mixing ratios were assessed with the model instead of calculated using its sensitivity.

The reviewer is right that, in principle, propene VMR could be assessed from basic reaction kinetics according to Holzinger et al. (2019). However, the $C_3H_6H^+$ ion is also a prominent fragment originating from several compounds (e.g. isoprene) and therefore we used modelled concentrations. In addition, propene was not detectable by the VOCUS PTR as a low mass filter was used.
* * *
Page 8, Line 242: Please provide output from the unweighted linear regression (e.g., correlation coefficient, p-value), and incorporate the output into your discussion on the agreement between $\sum k_i [VOC]_i$ and $k^{NO_3}$ measurements.

Done. We provided the correlation coefficient r of 0.95 (also denoted in Figure 2(b)) and now write (L245): A correlation coefficient of 0.95 underlines linearity of the whole data set despite increased scatter caused by the unfavourable conditions during type 2 experiments.
* * *
Page 13, Line 388–395: Please merge the model output (with $k_{wall} = 0 \ s^{-1}$) in Figure S3 to Figure 9, this could help the reader better visualize the effect of introducing the NO₃ and N₂O₅ wall loss.

Done. Figure 9 has been changed accordingly. In order to preserve legibility of the NO₃ and N₂O₅ measurements after implementation of the model output in (old) Fig. S3 to Fig. 9 the order and sizes of the panels were changed. Old Fig. S3 has been removed. The caption of Fig. 9 now reads:

Figure 9: O₃, NO₂, NO₃, N₂O₅ and isoprene mixing ratios and NO₃ reactivity on 2ⁿᵈ August (black). The grey shaded area symbolizes the overall uncertainty associated with each measurement. Orange circles denote the reactivity obtained using Eq.(3). The results of the numerical simulation using MCM v.3.3.1 with NO₃ and N₂O₅ wall loss rates set to 0 s⁻¹ (model 1) are shown by black lines. The model output with introduction of NO₃ and N₂O₅ wall loss rates of 0.016 s⁻¹ and 3.3 x 10⁻⁴ s⁻¹ respectively for each of the reactants is shown by a red line (model 2), whereas the blue line (model 3) shows the result of model 2 with the rate coefficient for reaction between NO₃ and RO₂ set to $4.6 \times 10^{-12}$ cm³molecule⁻¹s⁻¹, which is twice the value estimated by the MCM.

Please discuss more about the effect of omitting the NO₃ and N₂O₅ wall loss and its cause of large discrepancies between the measurement and model simulation in NO₃, N₂O₅, and isoprene mixing ratios.

The changes in Figure 9 and this comment necessitated to following changes in the manuscript text (L396-410):

We examined the effect of introducing the NO₃ and N₂O₅ wall loss rate constants calculated as described above into the chemical scheme used in the box model ( MCM v3.3.1). The results from three different model outputs for the experiment on the 2ⁿᵈ August are summarised in Fig. 9 which compares simulated and measured mixing ratios of NO₃, N₂O₅, NO₂, O₃ and isoprene (following its addition at 11:00) as well as the measured and non-steady-state NO₃ reactivities $k^{NO_3}$ and $k_{nss}^{NO3}$. The omission of NO₃/N₂O₅ wall losses (Model 1) results in simulated NO₃ and N₂O₅ mixing ratios up to 1400

and 1600 pptv during the isoprene-free period, which exceed measurements by factors of 4-8. This is because the only loss process for these species in this phase is the dilution rate that is two orders of magnitude lower than the estimated wall loss rates. Such high amounts of $NO_3$/$N_2O_5$ in the ppbv range result in rapid depletion of nearly half of the total injected isoprene within the first minute which is why Model 1 cannot describe the measurements either before or after the injection. Model 2 (red lines) includes the estimated wall loss rates and reproduces the measurements more accurately: The $NO_2$ and $O_3$ mixing ratios are accurately simulated. Furthermore, $NO_3$ and $N_2O_5$ mixing ratios that are only 10 to 30% higher than those measured and therefore $NO_3$ reactivities lower than $k_{nss}^{NO3}$ (orange circles) are predicted.

Please discuss how is the first-order wall loss rate for $O_3$, $H_2O_2$, HO, HONO and $HNO_3$ derived in Table S1.

The wall loss rates were derived as previously described (Richter, 2007). Compared to losses by dilution and reactions, this is a very minor sink that does not have a significant impact on the fate of $NO_3$.

The appropriate reference was added to table S1.
* * *
Page 13, Line 391: "and isoprene (following its addition at 10:50)": from Figure 9 and Figure S2, $NO_2$ appeared to be injected at 10:50, isoprene appeared to be injected at 11:00, please clarify.

Correct. $NO_2$ was injected at 10:50 and isoprene at 11:00 UTC.

We corrected this in the manuscript.
* * *
Page 22, Figure 2(b): To better aid visual inspection of the dataset, please set the aspect ratio of x:y to 1:1, add grid to x-axis and y-axis, add border to the legend (not shown in the demo below). See Figure 2(b).

Done. Figure 2(b) has been changed accordingly.

**2. Additional changes**

**L423:** Optimum agreement irrespective of uncertainties would be achieved with a value of 9.2 x 10$^{-12}$ cm³molecule$^{-1}$s$^{-1}$ for $k_{RO_2+NO_3}$ (i.e. a factor of 4 higher than in MCM) which is demonstrated in a comparable experiment under dry conditions on the 10$^{th}$ August (see Fig. S4 in the supplement).

**L443,483**: "within uncertainties" added

**Caption Fig. S4:** The results of the numerical simulation using MCM v.3.3.1 (with $NO_3$ and $N_2O_5$ wall loss rate of 0.016 s$^{-1}$ and 3.3 x 10$^{-4}$ s$^{-1}$ respectively) for each of the reactants is shown by a red line, whereas the blue line shows the result of the same model with  = 9.2 x 10$^{-12}$ cm³molecule$^{-1}$s$^{-1}$

**3. References**

Richter, C.A.: Ozone Production in the Atmosphere Simulation Chamber SAPHIR, Ph.D. thesis, Forschungszentrum Jülich GmbH, University of Köln, http://juser.fz-juelich.de/record/62596/files/Energie&Umwelt_02.pdf, 2007. (pp 37, 123)

Holzinger, R., Acton, W. J. F., Bloss, W. J., Breitenlechner, M., Crilley, L. R., Dusanter, S., Gonin, M., Gros, V., Keutsch, F. N., Kiendler-Scharr, A., Kramer, L. J., Krechmer, J. E., Languille, B., Locoge, N., Lopez-Hilfiker, F., Materić, D., Moreno, S., Nemitz, E., Quéléver, L. L. J., Sarda Esteve, R., Sauvage, S., Schallhart, S., Sommariva, R., Tillmann, R., Wedel, S., Worton, D. R., Xu, K., and Zaytsev, A.: Validity and limitations of simple reaction kinetics to calculate concentrations of organic compounds from ion counts in PTR-MS, Atmos. Meas. Tech., 12, 6193–6208, https://doi.org/10.5194/amt-12-6193-2019, 2019.

**Reply to Anonymous Referee #2**

*In the following, the referee's comments are reproduced (black) along with our replies (blue) and changes made to the text (red) in the revised manuscript.*

The authors report on studies of NO3 reactivity during 'nighttime' experiments in the SAPHIR chamber, with a primary focus on isoprene chemistry. An FT-CRDS system is used to determine the NO3 reactivity with respect to stable products in the chamber, while a box model analysis is used to assess additional NO3 losses (reaction with peroxy radicals, chamber wall losses) not determined by the FT-CRDS system. Among the key findings are the following: the FT-CRDS accurately measures the NO3 reactivity towards isoprene, and functions well under the conditions studied; stable products of the NO3/isoprene chemistry do not contribute significantly to NO3 reactivity; the generic (and highly uncertain) RO2 + NO3 rate coefficient may be a factor of two or more higher than current estimates. Overall, this is a very solid paper that certainly is publishable in ACP. The paper is well written, and assumptions and uncertainties in the measurements are generally presented in detail. A few questions and suggestions are presented below for the authors to consider.

We thank the referee for the positive evaluation of our manuscript and the useful comments.

**1. Referee's comments**

There are assumptions and caveats associated with equation (1), line 220 – Could there be significant reaction products that the PTR-MS is unable to detect? Could some products not make it into the flow tube for detection by the k(NO3) instrument? NO3 losses due to chamber walls and radicals are not measured by the k(NO3) instrument. Most (or maybe all) of these are dealt with at different points in the manuscript, but a clear statement or two delineating these at this point might be helpful to the reader.

This is indeed necessary for validity of Eq. (1). We now write (L219):

The VOC contribution to the $NO_3$ reactivity is the summed, first-order loss rate coefficient attributed to all non-radical VOCs present in the chamber that can be transported to the FT-CRDS according to Eq. (1):

Can the authors be more quantitative regarding the b-caryophyllene expt (Fig 3a)? -e.g., What is its expected lifetime? The k($NO_3$) instrument is clearly not seeing the full impact of the stated addition of 2 ppbv b-caryophyllene.

Assuming 120 ppbv of $O_3$ and a rate constant of $1.2 \times 10^{-14}$ cm$^3$molecule$^{-1}$s$^{-1}$ (298 K, IUAPC) for the reaction between β-caryophyllene and $O_3$ leads to a loss rate of 0.035 s$^{-1}$. Neglecting secondary oxidation, only 11 pptv of β-caryophyllene (resulting in $k^{NO_3}$ of 0.005 s$^{-1}$, which is the setup's LOD) are left after 150 s. The instrument was zeroing until a couple of minutes after the injection of β-caryophyllene and thus detected only the last residues of this sticky monoterpene. We add this point to the manuscript and now write (L258):

The instrument was zeroing until shortly after the injection of this terpene.  As the lifetime of β-caryophyllene is extremely short in the chamber under the given conditions (~ 150 s), only the small fraction of unreacted β-caryophyllene  contribute to the $k^{NO_3}$ signal observed after 08:40 UTC.

Line 265 / Fig 3b: Isoprene loss here is due to reaction with O3, I assume (maybe also OH formed in the ozonolysis)? Does the agreement noted between the k(NO3) instrument and the k[isoprene] calculation imply that major isoprene ozonolysis products are also comparatively unreactive towards NO3? (Also, a minor detail, but the isoprene decay seems more rapid than would be implied by the O3 concentration given?)

Correct, the isoprene loss is mainly caused by ozonolysis but also by dilution during the first three hours between 06:50 and 09:50 UTC. Using stated initial concentrations and rate coefficients at 298 K (IUPAC, 2019) calculated losses are as follows:

$$k_{loss}(isoprene) = k_{ozonolysis} + k_{dilution} = [O_3] * k_{O_3+isoprene} + k_{dilution}$$
$$= (3.11 * 10^{-5} + 1.5 * 10^{-5})s^{-1} = 4.61 * 10^{-5}s^{-1}$$

$$[isoprene](3\ h) \approx [Isoprene]_0 * \exp(-k_{loss}(isoprene) * 10800\ s)$$
$$\approx 4\ ppbv * \exp(-4.61 * 10^{-5}s^{-1} * 10800\ s) \approx 2.4\ ppbv$$

After 3 hours 2.4 ppbv of isoprene causing an $NO_3$ reactivity of 0.038 s$^{-1}$ which is in good agreement with the measurement. The sudden decrease in isoprene (and $k^{NO_3}$) after 09:50 UTC is caused by an increase of the dilution flow by a factor of 10 in scope of a humidification process.

We agree, the good agreement between the FT-CRDS measurement and k[isoprene] suggests a neglectable contribution of products from the ozonolysis. Given the low reactivity of stable ozonolysis products (e.g. MACR, MVK, formaldehyde) and the non-detection of radicals/Criegée intermediates this seems to be a valid conclusion. We include these aspects to the manuscript (L267):

Isoprene depletion is dominated by ozonolysis at this phase, whereas the sudden drop in $k^{NO_3}$ is caused by an increased dilution flow during humidification of the chamber around 10:00 UTC. The absence of $NO_2$ results in a more accurate, less scattered measurement of $k^{NO_3}$ and underscores the reliability of the measurement under favourable conditions. All of the observed reactivity can be assigned to isoprene that was injected at 06:52 UTC. This implies that stable secondary oxidation of products from isoprene ozonolysis (such as formaldehyde, MACR, MVK) are insignificant for $k^{NO_3}$ which is consistent with the low rate coefficients (e.g. $k_{MACR+NO_3} = 3.4\ x\ 10^{-15} cm^3 molecule^{-1}\ s^{-1}$ as highest of the three; IUPAC, 2019).

Line 312 or so - It should be noted here that NC4CHO is only one of many products that can be formed.

Correction made, we now write (L319):

One of several  major, stable oxidation products according to MCM is an organic nitrate with aldehyde functionality ($O_2NOC_4H_6CHO$, NC4CHO).

In Figure 9, it is not clear to me that the increased $RO_2$ + $NO_3$ rate coefficient improves the model/measured $NO_3$ comparison?

This statement referred to the very first phase after the isoprene injection, but we agree that in the last phase of the experiment (old) model 2 shows a worse agreement with the $NO_3$ measurement than (old) model 1. We now write (L419):

The higher rate coefficient for reaction of $NO_3$ with $RO_2$ would be sufficient to  explain the observed discrepancy between the overall reactivity $k_{nss}^{NO_3}$ and $k^{NO_3}$ within the uncertainties associated with the analysis.

**2. Additional changes**

**L423:** Optimum agreement irrespective of uncertainties would be achieved with a value of 9.2 x 10$^{-12}$ cm³molecule$^{-1}$s$^{-1}$ for $k_{\mathrm{RO_2+NO_3}}$ (i.e. a factor of 4 higher than in MCM) which is demonstrated in a comparable experiment under dry conditions on the 10$^{th}$ August (see Fig. S4 in the supplement).

**L443,483:** "within uncertainties" added

[revised manuscript text omitted]

During the experiment of the 23$^{rd}$ August (lower panel, Fig. 3b), only isoprene and ozone were present in the chamber for the first 4 hours. Isoprene depletion is dominated by ozonolysis at this phase, whereas the sudden drop in $k^{NO_3}$ is caused by an increased dilution flow during humidification of the chamber around 10:00 UTC. The absence of NO₂ results in a more accurate, less scattered measurement of $k^{NO_3}$ and underscores the reliability of the measurement under favourable conditions.

270 All of the observed reactivity can be assigned to isoprene that was injected at 06:52 UTC. This implies that stable secondary oxidation of products from isoprene ozonolysis (such as formaldehyde, MACR, MVK) are insignificant for $k^{NO_3}$ which is consistent with the low rate coefficients (e.g. $k_{MACR+NO_3} = 3.4 \; x \; 10^{-15} cm^3 molecule^{-1} \; s^{-1}$ as highest of the three; IUPAC, 2019).

[revised manuscript text omitted]

*Correspondence to*: John N. Crowley (john.crowley@mpic.de)

**Supplementary Information**

**Box-Model**

**Table S1: Reactions, rate coefficients and definitions in the model used for analysis. The isoprene oxidation scheme until the 3[rd] / 4[th] generation from the Master Chemical Mechansism (MCM) version 3.3.1 is used (Jenkin et al., 2015). Any change from MCMv3.3.1 is annotated.**

| Reaction | Reaction constant | Annotations |
|---|---|---|
| **NOx chemistry** | | |
| N2O5 → NO3 + NO2 | ((1.3e-3*(T/300)@-3.5*exp(-11000/T))*M* (9.7e14*(T/300)@0.1*exp(-11080/T)))/((1.3e-3* (T/300)@-3.5*exp(-11000/T))*M+(9.7e14*(T/300)@0.1* exp(-11080/T)))*10@(log10(0.35)/(1+(log10((1.3e- 3*(T/300)@-3.5 *exp(-11000/T))*M/(9.7e14*(T/300)@0.1*exp(-11080/T))) /(0.75-1.27*log10(0.35)))@2)) | |
| NO2 + NO3 → N2O5 | ((3.6e-30*(T/300)@-4.1)*M*(1.9e-12*(T/300)@0.2)) /((3.6e-30*(T/300)@-4.1)*M+(1.9e-12*(T/300)@0.2))* 10@(log10(0.35)/(1+(log10((3.6e-30*(T/300)@-4.1)* M/(1.9e-12*(T/300)@0.2))/(0.75-1.27*log10(0.35)))@2)) | |
| NO + O3 → NO2 + O2 | 1.8E-11*exp(110/T) | |
| NO2 + O3 → NO3 + O2 | 1.4E-13 * exp (-2470/T) | |
| NO + O3 → NO2 + O2 | 2.07E-12 * exp (-1400/T) | |
| NO3 + CO → | 4E-19 | Hjorth et al., 1986 |
| OH + NO2 →HNO3 | ((3.2e-30*(T/300)@-4.5)*M*(3.0e-11))/ ((3.2e-30*(T/300)@-4.5)*M+(3.0e-11))*10@(log10(0.41)/ (1+(log10((3.2e-30*(T/300)@-4.5)*M/(3.0e-11))/ (0.75-1.27*log10(0.41)))@2)) | |
| OH + NO3 →HO2 + NO2 | 2E-11 | |
| HO2 + NO3 → OH + NO2 | 4E-12 | |
| OH + NO → HONO | ((7.4e-31*(T/300)@-2.4)*M*(3.3e-11*(T/300)@-0.3))/ ((7.4e-31*(T/300)@-2.4)*M+(3.3e-11*(T/300)@-0.3))* 10@(log10(0.81)/(1+(log10((7.4e-31*(T/300)@-2.4)*M/ (3.3e-11*(T/300)@-0.3))/(0.75-1.27*log10(0.81)))@2)) | |
| HO2 + NO → OH + NO2 | 3.45E-12*exp(270/T) | |
| HO2 + NO2 → HO2NO2 | ((1.4e-31*(T/300)@-3.1)*M*(4.0e-12))/ ((1.4e-31*(T/300)@-3.1)*M+(4.0e-12))*10@(log10(0.4)/ (1+(log10((1.4e-31*(T/300)@-3.1)*M/(4.0e-12))/ (0.75-1.27*log10(0.4)))@2)) | |
| HO2NO2 + OH → NO2 | 3.2e-13*EXP(690/T) | |
| HO2NO2 → HO2 + NO2 | ((4.1e-5*exp(-10650/T))*M*(6.0e15*exp(-11170/T)))/ ((4.1e-5*exp(-10650/T))*M+(6.0e15*exp(-11170/T)))* 10@(log10(0.4)/(1+(log10((4.1e-5*exp(-10650/T))*M/ (6.0e15*exp(-11170/T)))/(0.75-1.27*log10(0.4)))@2)) | |

| | | |
|---|---|---|
| OH + HONO → NO2 | 2.5e-12*EXP(260/T) | |
| OH + HNO3 → NO3 | 2.40E-14*EXP(460/T) + ((6.50E-34*EXP(1335/T)*M)/ (1+(6.50E-34*EXP(1335/T)*M/2.70E-17*EXP(2199/T)))) | |
| **HOx chemistry** | | |
| OH + O3 → HO2 | 1.70E-12*EXP(-940/T) | |
| HO2 + O3 → OH | 2.03E-16*(T/300)@4.57*EXP(693/T) | |
| OH + HO2 → | 4.8E-11*EXP(250/T) | |
| HO2 + HO2 → H2O2 | 2.20E-13*(1+(1.40E-21*EXP(2200/T)*H2O))*EXP(600/T) | |
| OH + H2O2 → HO2 | 2.9E-12*exp(-160/T) | |
| OH + CO → HO2 | 1.44E-13*(1+(M/4.2E19)) | |
| **Primary oxidation of isoprene** | | |
| NO3 + C5H8 → NISOPO2 | 2.95E-12 * exp (-450/T) | IUPAC, 2019 |
| O3 + C5H8 → CH2OOE + MACR | 0.3 * 1.03E-14 * exp (-1995/T) | |
| O3 + C5H8 → CH2OOE + MVK | 0.2 * 1.03E-14 * exp (-1995/T) | |
| O3 + C5H8 → HCHO + MACROOA | 0.3 * 1.03E-14 * exp (-1995/T) | |
| O3 + C5H8 → HCHO + MVKOOA | 0.2 * 1.03E-14 * exp (-1995/T) | |
| OH + C5H8 → CISOPA | 0.288*2.7E-11 * exp (390/T) | |
| OH + C5H8 → CISOPC | 0.238*2.7E-11 * exp (390/T) | |
| OH + C5H8 → ISOP34O2 | 0.022*2.7E-11 * exp (390/T) | |
| OH + C5H8 → ME3BU3ECHO + HO2 | 0.02*2.7E-11 * exp (390/T) | |
| OH + C5H8 → PE4E2CO + HO2 | 0.042*2.7E-11 * exp (390/T) | |
| OH + C5H8 → TISOPA | 0.288*2.7E-11 * exp (390/T) | |
| OH + C5H8 → TISOPC | 0.102*2.7E-11 * exp (390/T) | |
| **Secondary oxidation (1st generation)** | | |
| NISOPO2 + HO2 → NISOPOOH | 0.706*2.91E-13 * EXP(1300/T) | |
| NISOPO2 + NO3 → NISOPO + NO2 | 2.3E-12 | |
| NISOPO2 + RO2 → ISOPCNO3 | 0.2*1.3E-12 | |
| NISOPO2 + RO2 → NC4CHO | 0.2*1.3E-12 | |
| NISOPO2 + RO2 → NISOPO | 0.6*1.3E-12 | |
| CH2OOE → CH2OO | 0.22*1E6 | |
| CH2OOE → CO | 0.51*1E6 | |
| CH2OOE → HO2 + CO + OH | 0.27*1E6 | |
| MACR + NO3 → MACO3 + HNO3 | 3.4E-15 | |

| Reaction | Rate | |
|---|---|---|
| MACR + O3 → HCHO + MGLYOOB | 0.12*1.4E-15*EXP(-2100/T) | |
| MACR + O3 → MGLYOX + CH2OOG | 0.88*1.4E-15*EXP(-2100/T) | |
| MACR + OH → MACO3 | 0.45*8.0E-12*EXP(380/T) | |
| MACR + OH → MACRO2 | 0.47*8.0E-12*EXP(380/T) | |
| MACR + OH → MACROHO2 | 0.08*8.0E-12*EXP(380/T) | |
| MVK + O3 → MGLOOA + HCHO | 0.5*8.5E-16*EXP(-1520/T) | |
| MVK + O3 → MGLYOX + CH2OOB | 0.5*8.5E-16*EXP(-1520/T) | |
| MVK + OH → HVMKAO2 | 0.3*2.6E-12*EXP(610/T) | |
| MVK + OH → HMVKBO2 | 0.7*2.6E-12*EXP(610/T) | |
| HCHO + NO3 → HNO3 + CO + HO2 | 5.5E-16 | |
| HCHO + OH → HO2 + CO | 5.4E-12 * exp (135/T) | |
| MACROOA → C3H6 | 0.255*1E6 | |
| MACROOA → CH3CO3 + HCHO + HO2 | 0.255*1E6 | |
| MACROOA → MACROO | 0.22*1E6 | |
| MACROOA → OH + CO +CH3CO3 + HCHO | 0.27*1E6 | |
| MVKOOA → C3H6 | 0.255*1E6 | |
| MVKOOA → CH3O2 + HCHO + CO + HO2 | 0.255*1E6 | |
| MVKOOA → MVKOO | 0.22*1E6 | |
| MVKOOA → OH + MVKO2 | 0.27*1E6 | |
| CISOPA + O2 → CISOPAO2 | 3.5E-12 | |
| CISOPA + O2 → ISOPBO2 | 3E-12 | |
| CISOPC + O2 → CISOPCO2 | 2E-12 | |
| CISOPC + O2 → ISOPDO2 | 3.5E-12 | |
| ISOP34O2 + HO2 → ISOP34OOH | 2.91E-13 * EXP(1300/T) | |
| ISOP34O2 + NO3 → ISOP34O + NO2 | 2.3E-12 | |
| ISOP34O2 + RO2 → HC4CHO | 0.1*2.65E-12 | |
| ISOP34O2 + RO2 → ISOP34O | 0.8*2.65E-12 | |
| ISOP34O2 + RO2 → ISOPDOH | 0.1*2.65E-12 | |
| ME3BU3ECHO + NO3 → NC526O2 | 3.3E-13 | |

| | | |
|---|---|---|
| ME3BU3ECHO + O3 → CH2OOC + CO2C3CHO | 0.33*1.6E-17 | |
| ME3BU3ECHO + O3 → HCHO + CO2C3OOB | 0.67*1.6E-17 | |
| ME3BU3ECHO + OH → C530O2 | 0.712*7.3E-11 | |
| ME3BU3ECHO + OH → ME3BU3ECO3 | 0.288*7.3E-11 | |
| PE4E2CO + NO3 → NC51O2 | 1.2E-14 | |
| PE4E2CO + O3 → CH2OOB + CO2C3CHO | 0.43*1E-17 | |
| PE4E2CO + O3 → HCHO + CO2C3OOA | 0.57*1E-17 | |
| PE4E2CO + OH → C51O2 | 2.71E-11 | |
| TISOPA + O2 → ISOPAO2 | 2.5E-12*exp(-480/T) | |
| TISOPA + O2 → ISOPBO2 | 3E-12 | |
| TISOPC + O2 → ISOPCO2 | 2.5E-12*exp(-480/T) | |
| TISOPC + O2 → ISOPDO2 | 3.5E-12 | |
| **Secondary oxidation (2nd generation)** | | |
| NISOPOOH + OH → NC4CHO + OH | 1.03E-10 | |
| NISOPO + O2 → NC4CHO + HO2 | 2.50E-14*EXP(-300/T) | |
| ISOPCNO3 + OH → INCO2 | 1.12E-10 | |
| NC4CHO + NO3 → NC4CO3 + HNO3 | 4.25*1.4E-12*EXP(-1860/T) | |
| NC4CHO + OH → C510O2 | 0.52*4.16E-11 | |
| NC4CHO + OH → NC4CO3 | 0.48*4.16E-11 | |
| NC4CHO + O3 → NOA + GLYOOC | 0.5*2.4E-17 | |
| NC4CHO + O3 → GLYOX + NOAOOA | 0.5*2.4E-17 | |
| CH2OO + CO → HCHO | 1.2E-15 | |
| CH2OO + NO2 → HCHO + NO3 | 1E-15 | |
| MACO3 + NO3 → CH3C2H2O2 + NO2 | 1.74 * 2.3E-12 | |
| MACO3 + HO2 → CH3C2H2O2 | 0.44 * 5.2E-13*EXP(980/T) | |
| MACO3 + HO2 → | 0.66 5.2E-13*EXP(980/T) | |
| MACO3 + RO2 → CH3C2H2O2 | 0.7*1E-11 | |
| MACO3 + RO2 → | 0.3*1E-11 | |

| | | |
|---|---|---|
| MGLYOOB → MGLYOO | 0.18*1E6 | |
| MGLYOOB → OH + CO + CH3CO3 | 0.82*1E6 | |
| MGLYOX + NO3 → CH3CO3 + CO + HNO3 | 2.4*1.4E-12*EXP(-1860/T) | |
| MGLYOX + OH → CH3CO3 + CO | 1.9E-12*exp(575/T) | |
| CH2OOG → CH2OO | 0.37*1E6 | |
| CH2OOG → CO | 0.47*1E6 | |
| CH2OOG → HO2 + CO + OH | 0.16*1E6 | |
| MACRO2 + HO2 → MACROOH | 0.625*2.91E-13 * EXP(1300/T) | |
| MACRO2 + NO3 → MACRO + NO2 | 2.3E-12 | |
| MACRO2 + RO2 → ACETOL | 9.2E-14 | |
| MACROHO2 + HO2 → (MACROHOOH) | 0.625*2.91E-13 * EXP(1300/T) | |
| MACROHO2 + NO3 → MACROHO + NO2 | 2.3E-12 | |
| MACROHO2 + RO2 → (div) | 1.4E-12 | |
| MGLOOA → CH3CHO | 0.2*1E6 | |
| MGLOOA → OH + CO + CH3CO3 | 0.36*1E6 | |
| MGLOOA → CH3CO3 + HCHO + HO2 | 0.2*1E6 | |
| MGLOOA → MGLOO | 0.24*1E6 | |
| CH2OOB → CH2OO | 0.24*1E6 | |
| CH2OOB → CO | 0.4*1E6 | |
| CH2OOB → HO2 + CO + OH | 0.36*1E6 | |
| HMVKAO2 + HO2 → (HMVKAOOH) | 0.625*2.91E-13 * EXP(1300/T) | |
| HMVKAO2 + NO3 → NO2 + HMVKAO | 2.3E-12 | |
| HMVKAO2 + RO2 → (div) | 2E-12 | |
| HMVKBO2 + HO2 → (HMVKBOOH) | 0.625*2.91E-13 * EXP(1300/T) | |
| HMVKBO2 + NO3 → NO2 + HMVKBO | 2.3E-12 | |
| HMVKBO2 + RO2 → (div) | 8.8E-13 | |
| C3H6 + O3 → CH2OOB + CH3CHO | 0.5*5.5E-15*EXP(-1880/T) | |

| | |
|---|---|
| C3H6 + O3 → CH3CHOOA + HCHO | 0.5*5.5E-15*EXP(-1880/T) |
| C3H6 + NO3 → PRONO3AO2 | 0.35*4.6E-13*EXP(-1155/T) |
| C3H6 + NO3 → PRONO3BO2 | 0.65*4.6E-13*EXP(-1155/T) |
| C3H6 + OH → HYPROPO2 | 0.87* ((8e-27*(T/300)@-3.5)*M*(3.0e-11*(T/300)@-1))/ ((8e-27*(T/300)@-3.5)*M+(3.0e-11*(T/300)@-1))* 10@(log10(0.5)/(1+(log10((8e-27*(T/300)@-3.5)*M/ (3.0e-11*(T/300)@-1))/(0.75-1.27*log10(0.5)))@2)) |
| C3H6 + OH → IPROPOLO2 | 0.13* ((8e-27*(T/300)@-3.5)*M*(3.0e-11*(T/300)@-1))/ ((8e-27*(T/300)@-3.5)*M+(3.0e-11*(T/300)@-1))* 10@(log10(0.5)/(1+(log10((8e-27*(T/300)@-3.5)*M/ (3.0e-11*(T/300)@-1))/(0.75-1.27*log10(0.5)))@2)) |
| CH3CO3 + HO2 → CH3CO2H + O3 | 5.2E-13*EXP(980/T) |
| CH3CO3 + NO3 → NO2 + CH3O2 | 4E-12 |
| CH3CO3 + RO2 → CH3CO2H | 0.3*1E-11 |
| CH3CO3 + RO2 → CH3O2 | 0.7*1E-11 |
| MACROO + CO → MACR | 1.2e-15 |
| MACROO + NO2 → MACR + NO3 | 1E-15 |
| CH3O2 + HO2 → | 3.8E-13*EXP(780/T)*(1-1/(1+498*EXP(-1160/T))) |
| CH3O2 + HO2 → HCHO | 3.8E-13*EXP(780/T)*(1/(1+498*EXP(-1160/T))) |
| CH3O2 + NO3 → CH3O + NO2 | 1.2E-12 |
| CH3O2 + RO2 → CH3OH | 0.5* 2*1.03E-13*EXP(365/T)*0.5*(1-7.18*EXP(-885/T)) |
| CH3O2 + RO2 → HCHO | 0.5* 2*1.03E-13*EXP(365/T)*0.5*(1-7.18*EXP(-885/T)) |
| MVKOO + CO → MVK | 1.2E-15 |
| MVKOO + NO2 → MVK + NO3 | 1E-15 |
| MVKO2 + HO2 → (MVKOOH) | 0.625*2.91E-13 * EXP(1300/T) |
| MVKO2 + NO3 → NO2 | 2.3E-12 |
| MVKO2 + RO2 → (div) | 2E-12 |
| CISOPAO2 + HO2 → ISOPAOOH | 0.706*2.91E-13 * EXP(1300/T) |
| CISOPAO2 + NO3 → CISOPAO + NO2 | 2.3E-12 |
| CISOPAO2 → C536O2 | 0.5*2.20E10*EXP(-8174/T)*EXP(1.00E8/T@3) |
| CISOPAO2 → C5HPALD1 + HO2 | 0.5*2.20E10*EXP(-8174/T)*EXP(1.00E8/T@3) |
| CISOPAO2 → CISOPA | 5.22E15*EXP(-9838/T) |
| CISOPAO2 + RO2→ CISOPAO | 0.8*2.4E-12 |

| | |
|---|---|
| CISOPAO2 + RO2 → HC4ACHO | 0.1*2.4E-12 |
| CISOPAO2 + RO2 → ISOPAOH | 0.1*2.4E-12 |
| ISOPBO2 + HO2 → ISOPBOOH | 0.706*2.91E-13 * EXP(1300/T) |
| ISOPBO2 + NO3 → ISOPBO + NO2 | 2.3E-12 |
| ISOPBO2 + RO2 → ISOPBO | 0.8*8E-13 |
| ISOPBO2 + RO2 → ISOPBOH | 0.2*8E-13 |
| CISOPCO2 + HO2 → ISOPCOOH | 0.706*2.91E-13 * EXP(1300/T) |
| CISOPCO2 + NO3 → CISOPCO + NO2 | 2.3E-12 |
| CISOPCO2 → C537O2 | 0.5*2.20E10*EXP(-8174/T)*EXP(1.00E8/T@3) |
| CISOPCO2 → C5HPALD2 + HO2 | 0.5*2.20E10*EXP(-8174/T)*EXP(1.00E8/T@3) |
| CISOPCO2 → CISOPC | 3.06E15*EXP(-10254/T) |
| CISOPCO2 + RO2 → CISOPCO | 0.8*2E-12 |
| CISOPCO2 + RO2 → HC4CCHO | 0.2*2E-12 |
| CISOPCO2 + RO2 → ISOPAOH | 0.2*2E-12 |
| ISOPDO2 + HO2 → ISOPDOOH | 0.706*2.91E-13 * EXP(1300/T) |
| ISOPDO2 + NO3 → ISOPDO + NO2 | 2.3E-12 |
| ISOPDO2 + RO2 → ISOPDO | 0.8*2.9E-12 |
| ISOPDO2 + RO2 → HCOC5 | 0.1*2.9E-12 |
| ISOPDO2 + RO2 → ISOPDOH | 0.1*2.9E-12 |
| ISOP34OOH + OH → HC4CHO + OH | 9.73E-11 |
| ISOP34O → MACR + HCHO + HO2 | 1E6 |
| HC4CHO + OH → C58O2 | 0.829*1.04E-10 |
| HC4CHO + OH → HC4CO3 | 0.171*1.04E-10 |
| ISOPDOH + OH → HCOC5 + HO2 | 7.38E-11 |
| NC526O2 + NO3 → NO2 + | 2.3E-12 |
| NC526O2 + RO2 → | 9.20E-14 |
| CH2OOC → CH2OO | 0.18*1E6 |
| CH2OOC → HO2 + CO+ OH | 0.82*1E6 |
| CO2C3CHO + NO3 → HNO3 + CO2C3CO3 | 4* 1.4E-12*EXP(-1860/T) |

| | | |
|---|---|---|
| CO2C3CHO + OH → CO2C3CO3 | 7.15E-11 | |
| CO2C3OOB → C4CO2O2 + OH | 0.82*1E6 | |
| CO2C3OOB → CO2C3OO | 0.18*1E6 | |
| C530O2 + HO2 → | 0.706*2.91E-13 * EXP(1300/T) | |
| C530O2 + NO3 → NO2 + | 2.3E-12 | |
| C530O2 + RO2 → | 9.2E-14 | |
| ME3BU3ECO3 + HO2 → C45O2 + OH + NO2 | 0.44*1.4E-12*EXP(-1860/T) | |
| ME3BU3ECO3 + HO2 → | 0.56*2.91E-13 * EXP(1300/T) | |
| ME3BU3ECO + NO3 → C45O2 + NO2 | 1.6*2.3E-12 | |
| ME3BU3ECO3 + RO2 → C45O2 | 1E-11 | |
| NC510O2 + HO2 → | 0.625*2.91E-13 * EXP(1300/T) | |
| NC510O2 + NO3 → NO2 + | 2.3E-12 | |
| NC510O2 + RO2 → | 8.8E-12 | |
| CO2C3OOA → C4CO2O2 + OH | 0.36*1E6 | |
| CO2C3OOA → CH2COCH2O2 + HO2 | 0.2*1E6 | |
| CO2C3OOA → CH2COCH3 | 0.2*1E6 | |
| CO2C3OOA → CO2C3OO | 0.24*1E6 | |
| C51O2 + HO2 → | 0.706*2.91E-13 * EXP(1300/T) | |
| C51O2 + NO3 → NO2 + | 2.3E-12 | |
| ISOPAO2 + HO2 → ISOPAOOH | 0.706*2.91E-13 * EXP(1300/T) | |
| ISOPAO2 + NO3 → NO2 + ISOPAO | 2.3E-12 | |
| ISOPAO2 + RO2 → HC4ACHO | 0.1*2.4E-12 | |
| ISOPAO2 + RO2 → ISOPAO | 0.8*2.4E-12 | |
| ISOPAO2 + RO2 → ISOPAOH | 0.1*2.4E-12 | |
| ISOPCO2 + HO2 → ISOPCOOH | 0.706*2.91E-13 * EXP(1300/T) | |
| ISOPCO2 + NO3 → NO2 + ISOPCO | 2.3E-12 | |
| ISOPCO2 + RO2 → HC4CCHO | 0.1*2E-12 | |
| ISOPCO2 + RO2 → ISOPAOH | 0.1*2E-12 | |
| ISOPCO2 + RO2 → ISOPCO | 0.8*2E12 | |
| **Secondary oxidation (3rd + generation)** | | |
| INCO2 + HO2 → | 0.706*2.91E-13 * EXP(1300/T) | |
| INCO2 + NO3 → NO2 + | 2.3E-12 | |
| INCO2 + RO2 → | 2.9E-12 | |

| | | |
|---|---|---|
| NC4CO3 + HO2 → NOA + CO+ HO2 + OH | 0.44*5.2E-13*EXP(980/T) | |
| NC4CO3 + HO2 → | 0.66*5.2E-13*EXP(980/T) | |
| NC4CO3 + NO3 → NOA + CO + HO2 + NO2 | 1.74*2.3E-12 | |
| NC4CO3 + RO2 → | 0.3*1E-11 | |
| NC4CO3 + RO2 → NOA + HO2 + CO | 0.7*1E-11 | |
| NOA + OH → MGLYOX + NO2 | 1.3E-13 | |
| C510O2 + HO2 → | 0.706*2.91E-13 * EXP(1300/T) | |
| C510O2 + NO3 → NO2 | 2.3E-12 | |
| C510O2 + RO2 → | 9.2E-14 | |
| GLYOOC → GLYOO | 0.11*1E6 | |
| GLYOOC → OH + HO2 + CO + CO | 0.89*1E6 | |
| GLYOO + NO2 → GLYOX + NO3 | 1E-15 | |
| NOAOOA → NOAOO | 0.11*1E6 | |
| NOAOOA → OH + NO2 + MGLYOX | 0.89*1E6 | |
| NOAOO + NO2 → NOA + NO3 | 1E-15 | |
| CH3C2H2O2 → CH3CO3 + HCHO | 0.35*1E6 | |
| CH3C2H2O2 → HCHO + CH3O2 + CO | 0.65*1E6 | |
| MGLYOO + NO2 → MGLYOX + NO3 | 1E-15 | |
| MACROOH + OH → ACETOL + CO + OH | 3.77E-11 | |
| MACRO → ACETOL + CO+ HO2 | 1E6 | |
| MACROHO → MGLYOX + HCHO + HO2 | 1E6 | |
| MGLOO + NO2 → MGLYOX + NO3 | 1E-15 | |
| HMVKAO → MGLYOX + HCHO + HO2 | 1E6 | |
| HMVKBO → CH3CO3 + HOCH2CHO | 1E6 | |
| CH3CHOOA → CH3CHOO | 0.24*1E6 | |

| | | |
|---|---|---|
| CH3CHOOA ➔ CH3O2 + CO + OH | 0.36*1E6 | |
| CH3CHOOA ➔ CH3O2 + HO2 | 0.2*1E6 | |
| CH3CHOOA ➔ | 0.2*1E6 | |
| CH3CHOO+ CO ➔ CH3CHO | 1.2E-15 | |
| CH3CHOO + NO2 ➔ CH3CHO + NO3 | 1E-15 | |
| PRONO3AO2 + HO2 ➔ | 0.520*2.91E-13 * EXP(1300/T) | |
| PRONO3AO2 + NO3 ➔ NO2 + | 2.3E-12 | |
| PRONO3AO2 + RO2 ➔ | 0.2*6E-13 | |
| PRONO3BO2 + HO2 ➔ | 0.520*2.91E-13 * EXP(1300/T) | |
| PRONO3BO2 + NO3 ➔ NO2 + | 2.3E-12 | |
| PRONO3BO2 + RO2 ➔ | 0.2*4E-14 | |
| HYPROPO2 + HO2 ➔ | 0.520*2.91E-13 * EXP(1300/T) | |
| HYPROPO2 + NO3 ➔ NO2 + | 2.3E-12 | |
| HYPROPO2 + RO2 ➔ | 8.8E-13 | |
| IPROPOLO2 + HO2 ➔ | 0.520*2.91E-13 * EXP(1300/T) | |
| IPROPOLO2 + NO3 ➔ NO2 + | 2.3E-12 | |
| IPROPOLO2 + RO2 ➔ | 2E-12 | |
| MVKOOH + OH ➔ VGLYOX | 2.55E-11 | |
| MVKOOH + OH ➔ MVKO2 | 1.90E-12*EXP(190/T) | |
| VGLYOX + NO3 ➔ | 2.0*1.4E-12*EXP(-1860/T) | |
| CH3CO2H + OH ➔ CH3O2 | 8E-13 | |
| ISOPAOOH + OH ➔ HC4ACHO | 0.05*1.54E-10 | |
| ISOPAOOH + OH ➔ IEPOXA + OH | 0.93*1.54E-10 | |
| ISOPAOOH + OH ➔ ISOPAO2 | 0.02*1.54E-10 | |
| HC4ACHO + NO3 ➔ HC4ACO3 + HNO3 | 4.25*1.4E-12*EXP(-1860/T) | |
| HC4ACHO + O3 ➔ ACETOL + GLYOX | 0.5*2.4E-17 | |
| HC4ACHO + O3 ➔ CO + | 0.5*2.4E-17 | |
| HC4ACHO + OH ➔ C58O2 | 0.52*4.52E-11 | |
| HC4ACHO + OH ➔ HC4ACO3 | 0.49*4.52E-11 | |
| C58O2 + HO2 ➔ | 0.706*2.91E-13 * EXP(1300/T) | |
| C58O2 + NO3 ➔ NO2 + | 2.3E-12 | |
| C58O2 + RO2 ➔ | 9.2E-14 | |
| HC4ACO3 + HO2 ➔ | 5.2E-13*EXP(980/T) | |
| HC4ACO3 + NO3 ➔ NO2 + | 1.74*2.3E-12 | |
| HC4ACO3 + RO2 ➔ | 1E-11 | |

| | | |
|---|---|---|
| HC4ACO3 ➝ HO2 + | 2.20E10*EXP(-8174/T)*EXP(1.00E8/T@3) | |
| CISOPAO ➝ C526O2 | 0.19*1E6 | |
| CISOPAO ➝ HC4CCHO + HO2 | 0.63*1E6 | |
| CISOPAO ➝ HO2 + M3F | 0.18*1E6 | |
| C526O2 + HO2 ➝ | 0.706*2.91E-13 * EXP(1300/T) | |
| C526O2 + NO3 ➝ NO2 + | 2.3E-12 | |
| C526O2 + RO2 ➝ | 9.20E-14 | |
| C526O2 ➝ CO + OH | 3.00E7*EXP(-5300/T) | |
| M3F + NO3 ➝ NO2 + | 1.9E-11 | |
| M3F + O3 ➝ | 2E-17 | |
| M3F + OH ➝ HO2 + | 9E-11 | |
| C536O2 + HO2 ➝ | 0.706*2.91E-13 * EXP(1300/T) | |
| C536O2 + NO3 ➝ NO2 + | 2.3E-12 | |
| C536O2 + RO2 ➝ | 9.20E-14 | |
| C536O2 ➝ CO + OH | 3.00E7*EXP(-5300/T) | |
| C5HPALD1 + NO3 ➝ OH + HNO3 + | 4.25*1.4E-12*EXP(-1860/T) | |
| C5HPALD1 + O3 ➝ MGLYOOA | 0.73*2.4E-17 | |
| C5HPALD1 + O3 ➝ MGLYOX | 0.27*2.4E-17 | |
| MGLYOOA ➝ MGLYOO | 0.11*1E6 | |
| MGLYOOA ➝ CH3CO3 + OH +CO | 0.89*1E6 | |
| C5HPALD1 + OH ➝ OH + | 5.2E-11 | |
| ISOPAOH + OH ➝ HC4ACHO+ HO2 | 0.5*9.3E-11 | |
| ISOPAOH + OH ➝ HC4CCHO + HO2 | 0.5*9.3E-11 | |
| HC4CCHO + NO3 ➝ HC4CCO3 + HNO3 | 4.25*1.4E-12*EXP(-1860/T) | |
| HC4CCHO + O3 ➝ | 2.4E-17 | |
| HC4CCHO + OH ➝ C57O2 | 0.52*4.52E-11 | |
| HC4CCHO + OH ➝ HC4CCO3 | 0.48*4.52E-11 | |
| HC4CCO3 + HO2 ➝ | 5.2E-13*EXP(980/T) | |
| HC4CCO3 + NO3 ➝ NO2 + | 1.74*2.3E-12 | |
| HC4CCO3 + RO2 ➝ | 1E-11 | |
| C57O2 + HO2 ➝ | 0.706*2.91E-13 * EXP(1300/T) | |
| C57O2 + NO3 ➝ NO2 + | 2.3E-12 | |
| C57O2 + RO2 ➝ | 9.20E-14 | |
| ISOPBOOH + OH ➝ IEPOXB + OH | 0.92*5E-11 | |

| | | |
|---|---|---|
| ISOPBOOH + OH → ISOPBO2 | 0.08*5E-11 | |
| IEPOXB + OH → IEB1O2 | 0.5*9.05E-12 | |
| IEPOXB + OH → IEB2O2 | 0.5*9.05E-12 | |
| IEB1O2 + HO2 → | 0.706*2.91E-13 * EXP(1300/T) | |
| IEB1O2 + NO3 → NO2 + | 2.3E-12 | |
| IEB1O2 + RO2 → | 9.20E-14 | |
| IEB1O2 + HO2 → | 0.706*2.91E-13 * EXP(1300/T) | |
| IEB1O2 + NO3 → NO2 + | 2.3E-12 | |
| IEB1O2 + RO2 → | 8.8E-13 | |
| ISOPBO → MVK + HCHO + HO2 | 1E6 | |
| ISOPBOH + OH → ISOPBO | 3.85E-11 | |
| ISOPCOOH + OH → HC4CCHO + OH | 0.05*1.54E-10 | |
| ISOPCOOH + OH → IEPOXC + OH | 0.93*1.54E-10 | |
| ISOPCOOH + OH → ISOPCO2 | 0.02*1.54E-10 | |
| IEPOXC + OH → IEC1O2 | 0.719*1.5E-11 | |
| IEPOXC + OH → | 0.281*1.5E-11 | |
| IEC1O2 + HO2 → | 0.706*2.91E-13 * EXP(1300/T) | |
| IEC1O2 + NO3 → NO2 + | 2.3E-12 | |
| IEC1O2 + RO2 → | 9.2E-14 | |
| CISOPCO → C527O2 | 0.3*1E6 | |
| CISOPCO → HC4ACHO | 0.52*1E6 | |
| CISOPCO → HO2 + M3F | 0.18*1E6 | |
| C527O2 + HO2 → | 0.706*2.91E-13 * EXP(1300/T) | |
| C527O2 + NO3 → NO2 + | 2.3E-12 | |
| C527O2 + RO2 → | 8.8E-13 | |
| C527O2 → CO + OH | 3.00E7*EXP(-5300/T) | |
| C537O2 + HO2 → | 0.706*2.91E-13 * EXP(1300/T) | |
| C537O2 + NO3 → NO2 + | 2.3E-12 | |
| C537O2 + RO2 → | 9.2E-14 | |
| C537O2 → CO + OH | 3.00E7*EXP(-5300/T) | |
| C5HPALD2 + NO3 → OH + HNO3 + | 4.25*1.4E-12*EXP(-1860/T) | |
| C5HPALD2 + O3 → MGLYOOC | 0.73*2.4E-17 | |
| C5HPALD2 + O3 → MGLYOX | 0.27*2.4E-17 | |
| C5HPALD2 + OH → OH | 5.2E-11 | |
| ISOPAOH + OH → HC4ACHO + HO2 | 0.5*9.3E-11 | |

| | | |
|---|---|---|
| ISOPAOH + OH → HC4CCHO + HO2 | 0.5*9.3E-11 | |
| ISOPDOOH + OH → HCOC5 + OH | 0.22*1.15E-10 | |
| ISOPDOOH + OH → IEPOXB + OH | 0.75*1.15E-10 | |
| ISOPDOOH + ISOPDO2 | 0.03*1.15E-10 | |
| OH + HCOC5 → C59O2 | 3.81E-11 | |
| C59O2 + HO2 → | 0.706*2.91E-13 * EXP(1300/T) | |
| C59O2 + NO3 → NO2 + | 2.3E-12 | |
| C59O2 + RO2 → | 9.2E-14 | |
| ISOPDO → MACR + HCHO + HO2 | 1E6 | |
| ISOPDOH + OH → HCOC5 | 7.38E-11 | |
| HC4CO3 + HO2 → | 0.56*2.91E-13 * EXP(1300/T) | |
| HC4CO3 + HO2 → MACR + HO2 + OH | 0.44*2.91E-13 * EXP(1300/T) | |
| HC4CO3 + NO3 → MACR + HO2 + NO2 | 1.5*2.3E-12 | |
| HC4CO3 → MACR + HO2 | 1E-11 | |
| CO2C3CO3 + HO2 → CH3COCH2O2 | 0.44*2.91E-13 * EXP(1300/T) | |
| CO2C3CO3 + HO2 → | 0.56*2.91E-13 * EXP(1300/T) | |
| CO2C3CO3 + NO3 → CH3COCH2O2 + NO2 | 1.74*2.3E-12 | |
| CO2C3CO3 → CH3COCH2O2 | 1E-11 | |
| CH3COCH2O2 + HO2 → OH + | 0.15*1.36E-13*EXP(1250/T) | |
| CH3COCH2O2 + HO2 → | 0.85*1.36E-13*EXP(1250/T) | |
| CH3COCH2O2 + NO3 → NO2 + | 2.3E-12 | |
| CH3COCH2O2 + RO2 → ACETOL | 0.2* 2*(3.5E-13*8E-12)@0.5 | |
| CH3COCH2O2 + RO2 → | 0.6* 2*(3.5E-13*8E-12)@0.5 | |
| CH3COCH2O2 + RO2 → MGLYOX | 0.2* 2*(3.5E-13*8E-12)@0.5 | |
| CO2C3OO + CO → | 1.2E-15 | |
| CO2C3OO + NO2 → NO3 + | 1E-15 | |
| C4CO2O2 + HO2 → | 0.625*2.91E-13 * EXP(1300/T) | |
| C4CO2O2 + NO3 → NO2 + | 2.3E-12 | |
| C4CO2O2 + RO2 → | 8.8E-12 | |
| C45O2 + HO2 → | 0.625*2.91E-13 * EXP(1300/T) | |
| C45O2 + NO3 → NO2 + | 2.3E-12 | |

| Reaction | Rate | Notes |
|---|---|---|
| C45O2 + RO2 → | 1.3E-12 | |
| ISOPAO → C524O2 | 0.25*1E6 | |
| ISOPAO → HC4CHO + HO2 | 0.75*1E6 | |
| C524O2 + HO2 → | 0.706*2.91E-13 * EXP(1300/T) | |
| C5242 + NO3 → NO2 + | 2.3E-12 | |
| C5242 + RO2 → | 2.9E-12 | |
| ISOPCOOH + OH → HC4CCHO + OH | 0.05*1.54E-10 | |
| ISOPCOOH + OH → IEPOXC + OH | 0.93*1.54E-10 | |
| ISOPCOOH + ISOPCO2 | 0.02*1.54E-10 | |
| ISOPCO → HC4ACHO + HO2 | 0.75*1E6 | |
| ISOPCO → HC4CCHO + HO2 | 0.25*1E6 | |
| **β-caryophyllene** | | Jenkin et al., 2012 |
| BCARY + NO3 → NBCO2 | 1.9E-11 | |
| NBCO2 + NO3 → | 2.3E-12 | |
| BCARY + O3 → BCAOO | 0.435*1.2E-14 | |
| BCARY + O3 → BCBOO | 0.435*1.2E-14 | |
| BCARY + O3 → | 0.13*1.2E-14 | |
| BCAOO → BCSOZ | 8E1 | |
| BCBOO → BCSOZ | 1.2E2 | |
| **SAPHIR chamber** | | |
| Y + OH → HO2 | 1.44E-13*(1+(M/4.2E19)) | OH background reactivity; behaving like CO (Fuchs et al., 2013) |
| Z + wall → | 3.86E-6 | Wall loss for $O_3$, $H_2O_2$, $HO_2$, HONO and $HNO_3$ (Richter, 2007) |
| NO3 + wall → | 1.6E-3 | Wall loss NO3 |
| N2O5 + wall → | 3.3E-4 | Wall loss $N_2O_5$ |
| **Definitions** | | |
| RO2 | NISOPO2 + ISOP34O2 + CH3C2H2O2 + MACO3 + MACRO2 + MACROHO2 + CH3CO3 + HMVKAO2 + HMVKBO2 + CH3O2 + MVKO2 + CISOPAO2 + ISOPBO2 + CISOPCO2 + ISOPDO2 + NC526O2 + C530O2 + M3BU3ECO3 + C45O2 + NC51O2 + C51O2 + ISOPAO2 + ISOPCO2 + INCO2 + NC4CO3 + C510O2 + PRONO3AO2 + PRONO3BO2 + HYPROPO2 + IPROPOLO2 + C536O2 + C537O2 + INAO2 + C58O2 + HC4CO3 + CO2C3CO3 + CH3COCH2O2 + C4CO2O2 + C527O2 + C526O2 + HC4ACO3 HC4CCO3 + C57O2 + C59O2 + C524O2 | organic peroxides |
| kNO3_all | C5H8*2.95E-12*exp(450/T) + BCARY*1.9E-11 + C3H6*4.6E-13*exp(-1155/T) + (2.3E-12*(NISOPO2 + ISOPAO2 + | overall NO3 reactivity |

| | | |
|---|---|---|
| | ISOPBO2 + ISOPCO2 + ISOPDO2 + CH3C2H2O2 + MACO3 + MACRO2 + MACROHO2 + HMVKAO2 + HMVKBO2 + MVKO2 + INCO2 + CISOPAO + CISOPAO2 + (NC4CO3*1.74) + C510O2 + NBCO2 + PRONO3AO2 + PRONO3BO2 + HYPROPO2 + IPROPOLO2 + INAO2 + C524O2 + (HC4ACO3*1.74) + (1.6*HC4CO3) + C58O2 + INB1O2 + (HC4CCO3*2.74) + INDO2 + C57O2 + C59O2 + C51O2 + IEB1O2 + IEB2O2 + IEC1O2 + ISOP34O2 + CISOPCO2 + NC526O2 + C527O2 + C526O2 + C536O2 + C537O2 + C530O2 + C45O2 + 1.6*M3BU3ECO3 + INB2O2 + NC51O2 + 1.74*CO2C3CO3 + CH3COCH2O2 + C4CO2O2)) + (4E-12*CH3CO3) + (1.2E-12*CH3O2) + (HO2*4E-12) + (5.5E-16*HCHO) + (4E-19*CO) + 1.4E-12*EXP(-1860/T)*(NC4CHO*4.25 + HC4ACHO*4.25 + HC4CCHO*4.25 + 2.4*MGLYOX + 4*CO2C3CHO + 4.25*C5HPALD1 + 4.25*C5HPALD2 +2*VGLYOX) + 3.3E-13*ME3BU3ECHO + (M3F*1.9E-11) + (1.2E-14*PE4E2CO) | |
| kNO3_stable | C5H8*2.95E-12*exp(450/T) + BCARY*1.9E-11 + C3H6*4.6E-13*exp(-1155/T) + (5.5E-16*HCHO) + (4E-19*CO) + 1.4E-12*EXP(-1860/T)*(NC4CHO*4.25 + HC4ACHO*4.25 + HC4CCHO*4.25 + 2.4*MGLYOX + 4*CO2C3CHO + 4.25*C5HPALD1 + 4.25*C5HPALD2 +2*VGLYOX) + 3.3E-13*ME3BU3ECHO + (M3F*1.9E-11) + (1.2E-14*PE4E2CO) | NO$_3$ reactivity measurable by FT-CRDS |
| M | P*(3.24E16)*(298/T) | Total molecular concentration using measured pressure P in Torr and temperature T in K |

**Exemplary comparison of isoprene measurements**

[Figure]

**Figure S1: Amounts of isoprene during parts of the experiments on the 3rd and 6th August as measured by the two available PTR-ToF-MS instruments Vocus (black) and PTR1000 (red).**

**Comparison of $k^{OH}$ and $k^{NO_3}$**

During NO3ISOP, $k^{OH}$ was measured with an instrument based on laser photolysis – laser induced fluorescence (LP-LIF) (Hofzumahaus et al., 2009; Lou et al., 2010; Fuchs et al., 2017a; Fuchs et al., 2017b). Ambient air was passed at a flow rate of 19 L min$^{-1}$ through a flow tube and part of the air was drawn into an OH fluorescence detection cell. OH radicals were produced within a few nanoseconds in the flow tube by pulsed laser-photolysis of $O_3$ (at 266 nm) with subsequent reaction of $O(^1D)$ atoms with water vapour. OH concentration profiles were recorded by LIF, with $k^{OH}$ determined from the exponential decay constant after correction for diffusion / wall loss ($1.8 \pm 0.15$ s$^{-1}$). The time resolution of the $k^{OH}$ measurements was 90 s with a limit of detection of 0.5 s$^{-1}$. The resulting accuracy of $k^{OH}$ is (5-10) % $\pm 0.2$ s$^{-1}$ at NO mixing ratios below 20 ppbv.

Each isoprene injection results in an increase in reactivity of both OH and NO$_3$. Within the first few minutes after an isoprene injection, the contribution of secondary oxidation products to both $k^{NO_3}$ and $k^{OH}$ is negligible. Hence, the increase in the OH- and NO$_3$ reactivity ($\Delta k^{OH}$ and $\Delta k^{NO_3}$) directly after an isoprene injection scales with the amount of isoprene injected and the corresponding rate coefficient ($k_{NO_3+C_5H_8} = 6.5 \times 10^{-13}$ cm$^3$ molecule$^{-1}$ s$^{-1}$, $k_{OH+C_5H_8} = 1 \times 10^{-10}$ cm$^3$ molecule$^{-1}$ s$^{-1}$ at 298 K (IUPAC, 2019)). For any particular injection, both approaches should lead to similar isoprene concentrations as shown in Eq. S1.

$$[\text{Isoprene}] = \frac{\Delta k^{OH}}{k_{OH+C_5H_8}} = \frac{\Delta k^{NO_3}}{k_{NO_3+C_5H_8}} \tag{S1}$$

Figure S2 plots the isoprene mixing ratios derived from measurements of $\Delta k^{OH}$ versus those derived from $\Delta k^{NO_3}$. For experiments with isoprene mixing ratios below ~5 ppbv a slope of $0.88 \pm 0.11$ was obtained. During two injections, when high concentrations of isoprene (~11 and ~22 ppbv) were injected in the chamber, the $\Delta k^{OH}$ measurement returns isoprene mixing ratios that are significantly lower than those derived from $\Delta k^{NO_3}$ and the mixing ratio expected from the amount of isoprene injected. On these days, a combination of the low laser power and a small number of points to fit the (rapid) exponential decay mean that the OH reactivity must be considered a lower-limit.

[Figure]

**Figure S2:** Isoprene mixing ratios deduced from $\Delta k^{OH}$ against those from $\Delta k^{NO_3}$ under the usage of Eq. (S1) for isoprene injections of different experiments (days). The error bars denote the associated uncertainties in $\Delta k^{NO_3}$ (4-70%, Liebmann et al., 2017) and $k_{NO_3 + C_5H_8}$ (41% (IUPAC, 2019)) and $\Delta k^{OH}$ (10%, for [isoprene] < 5 ppbv) and $k_{OH + C_5H_8}$(15% (IUPAC, 2019)). The black line indicates the case of ideal 1:1 correlation, the red line shows an orthogonal linear regression (slope: 0.88 ± 0.11, intercept: 0.17 ± 0.23) for data points < 5 ppbv.

80

**Validity of the steady-state assumption**

85 The validity of the steady-state assumption was checked with the help of a correlation plot between the steady-state ($k_{SS}^{NO_3}$) and non-steady-state ($k_{nss}^{NO_3}$) reactivity as depicted in Fig. S3a. A slope close to 1 is found for most of the experiments. At injection points of $NO_2$ or at low reactivities larger differences are observed which are related to short-term perturbation of the equilibrium between $NO_3$ and $N_2O_5$ and deviation from steady-state.

90 Figure S3b compares $k_{ss}^{NO3}$ with $k_{nss}^{NO3}$ on the 2nd August. Between 9:00 and 11:00 UTC only $NO_2$ and $O_3$ were injected into chamber so that the influence of the chamber alone (reaction with the walls and the dilution flow) determines the $NO_3$ losses. As the $NO_3$ loss rate is low under these circumstances, nearly half an hour is necessary to achieve steady-state. This is confirmed by the difference between $k_{nss}^{NO3}$ and $k_{ss}^{NO3}$. Under the experimental conditions, the equilibrium between $NO_3$ and $N_2O_5$ is reached more rapidly than the steady state (Brown et al., 2003). Consequently, $k_{nss}^{NO3}$ acquires a constant value earlier

95 than $k_{ss}^{NO3}$. A reinjection of $NO_2$ at ~10:50 perturbs the stationary-state and therefore strongly affects $k_{ss}^{NO3}$ whereas $k_{nss}^{NO3}$ remains mostly unchanged. After the injection of isoprene the high $NO_3$-reactivity means that the steady-state assumption becomes valid, which leads to an agreement between the two methods.

[Figure]

100

(b)

[Figure]

**Figure S3: (a)** Steady-state $k_{SS}^{NO_3}$ and non-steady-state $k_{nss}^{NO_3}$ reactivities sorted by experiment. The dotted line through the origin with a slope of 1 represents perfect agreement. **(b)** Comparison between steady- (red) and non-steady-state (blue) reactivities on the experiment of the 2$^{nd}$ August. The respective uncertainties obtained from error propagation of the uncertainties in $k_2$ (15%; IUPAC, 2019) and the NO$_3$, NO$_2$ and O$_3$ mixing ratios (25%, 9% and 5%, respectively) are indicated by areas in the same colour of the data points.

[Figure]

**Figure S4:** $O_3$, $NO_2$, $NO_3$, $N_2O_5$ and isoprene mixing ratios as well as the $NO_3$ reactivity on the experiment of the $10^{th}$ August (black). The grey shaded area symbolizes the overall uncertainty associated with each measurement. Orange circles denote the non-steady-state reactivity obtained from Eq.(3). The results of the numerical simulation using MCM v.3.3.1 (with $NO_3$ and $N_2O_5$ wall loss rate of 0.016 $s^{-1}$ and 3.3 x $10^{-4}$ $s^{-1}$ respectively) for each of the reactants is shown by a red line, whereas the blue line shows the result of the same model with  ($k_{NO_3+RO_2}$ = **9.2** x $10^{-12}$ $cm^3$molecule$^{-1}$s$^{-1}$)